# Learning Robust Vision-Language Models from Natural Latent Spaces

**Zhangyun Wang**[*]
School of Computer Science
University of Auckland
zwna875@aucklanduni.ac.nz

**Ni Ding**
School of Computer Science
University of Auckland
ni.ding@auckland.ac.nz

**Aniket Mahanti**
School of Computer Science
University of Auckland
a.mahanti@auckland.ac.nz

## Abstract

Pre-trained vision-language models (VLMs) exhibit significant vulnerability to imperceptible adversarial perturbations. Current advanced defense strategies typically employ adversarial prompt tuning to improve the adversarial robustness of VLMs, which struggle to simultaneously maintain generalization across both natural and adversarial examples under different benchmarks and downstream tasks. We propose a collaborative adversarial prompt tuning (CoAPT) approach from pre-trained VLMs to target robust VLMs. Inspired by the image mask modeling, we adopt an improved real-time total variation algorithm to suppress and eliminate high-frequency details from images while preserving edge structures, thereby disrupting the adversarial perturbation space. Subsequently, guided by the high-level image and text representations in the latent space of the pre-trained VLMs, the corrupted natural features are restored while inheriting the superior generalization capability. Experiments on four benchmarks demonstrate that CoAPT achieves an excellent trade-off among natural generalization, adversarial robustness, and task-specific adaptation compared to state-of-the-art methods.

## 1 Introduction

Vision-language models (VLMs) such as CLIP[1] and ALBEF[2] have shown significant potential for application in multiple industry ecosystems in recent years. However, recent studies [3, 4] have revealed that VLMs exhibit a range of concerning vulnerabilities in real-world deployment. When confronted with distributional biases, adversarial samples, or semantic ambiguities, they often display reasoning biases that deviate from human cognition. As an increasing number of downstream applications built upon VLMs as foundational models emerge, the chain reactions triggered by the vulnerability of VLMs pose serious threats to the security and reliability of multimodal downstream tasks. In this paper, we holistically investigate the vulnerabilities of VLMs and their adversarial robustness, with a particular focus on the typical base model CLIP.

Current adversarial robustness strategies for VLMs primarily include model fine-tuning and adversarial prompt tuning. During adversarial training, model fine-tuning [5, 6] relearns the entire set of model parameters to adapt to adversarial examples. This process disrupts the natural data distribution captured by the pre-trained model, leading to a contradiction between robustness and generalization.

---

[*]Corresponding author: Zhangyun Wang (zwna875@aucklanduni.ac.nz)

39th Conference on Neural Information Processing Systems (NeurIPS 2025).

Adversarial prompt tuning [7, 8, 9] improves the robust adaptability of VLMs by guiding the pre-trained models to efficiently adapt to adversarial data distributions, without altering the pre-trained model parameters. Textual adversarial prompt tuning [10, 11] employs learnable prompts in the language branch to match and counteract adversarial attacks from the visual branch. In contrast, visual adversarial prompt tuning C-AVP[12] directly recognize and refine the adversarial images to allow the pre-trained models to make more accurate predictions. More promising multimodal adversarial prompt methods [7, 13, 14, 8] simultaneously introduce deep learnable prompts into the visual and language branches to achieve more comprehensive adversarial robustness. Although adversarial prompt tuning preserves the generalized feature representations of pre-trained VLMs, excessive reliance on in-distribution adversarial samples causes degradation of their natural generalization distribution during the adaptation process. Out-of-distribution (OOD) or unseen tasks further challenge the natural generalization and robustness of prompt-tuned VLMs[10].

Pre-trained VLMs retain generalizable knowledge for unseen tasks, while adversarial prompts can guide the shift of natural distribution toward adversarial-robust distributions or downstream task-specific distributions [15]. Therefore, we propose to leverage adversarial prompt tuning to identify a shared latent distribution that effectively balances natural generalization, adversarial robustness, and task-specific adaptation. Due to the inherent discrepancies among different distributions, directly training models with a mixture of natural and adversarial samples to fit the latent distribution leads to suboptimal solutions. Recent findings [16, 17] indicate that masked image modeling (MIM) enables models to learn more generalizable and robust representations, which significantly enhances their capacity to adapt to input distribution variations and improve fine-tuning performance in downstream vision tasks. The success of MIM is due to masked image input and image-level reconstruction objectives. However, this paradigm directs the model to pay more attention to high-frequency (HF) components where adversarial perturbations are concentrated, thus failing to effectively improve adversarial robustness [9]. We propose a collaborative adversarial prompt tuning (CoAPT) in which pre-trained CLIP collaborates with a target robust CLIP to address this issue. We convert the patch-level image masking from MIM to pixel-level image corruption for model inputs. An improved real-time total variation (TV) regularization method is employed to suppress the adversarial perturbation space by drastically smoothing the high-frequency details of the input images while preserving the image edge structures. To mitigate the cost of sacrificing natural high-frequency features, we shift the reconstruction objective from the pixel space of the target robust CLIP to the latent representation space of the natural CLIP. The corrupted natural detail features are restored under the guidance of high-level features of natural CLIP images and texts, thereby inheriting their excellent generalization ability. Overall, the fine-tuned adversarial prompts work in synergy with the frozen weights of the original pre-trained CLIP to support the target robust CLIP. They achieve a good balance between **(a)** improving adversarial robustness while maintaining natural performance on in-distribution tasks, and **(b)** maintaining natural generalization while enhancing the robust adaptability of the original VLMs on OOD or unseen tasks. Our contributions are threefold:

- We propose a novel paradigm for adversarial prompt tuning that learns robust CLIP from the latent space of natural CLIP. CoAPT weakens high-frequency details of input images to suppress the adversarial perturbation space. Guided by natural CLIP, corrupted generalization features are restored in the latent space. We introduce Rényi divergence to minimize the discrepancy between the similarity distributions of adversarial and natural examples.

- We design a real-time adaptive TV regularization method to efficiently suppress the perturbation space. It addresses the slow convergence and residual adversarial perturbations of traditional TV regularization by combining a spatially adaptive regularization strategy based on edge strength response and an accelerated gradient method with adaptive restart.

- An optimal trade-off among natural generalizability, adversarial robustness, and task-specific adaptation is achieved. Without benchmark-specific or dataset-specific hyperparameter tuning, we improve natural and adversarial robustness performance on 15 datasets across four benchmarks by an average of 9.83% and 24.16%, respectively.

## 2 Related Work

**Adversarial attacks on VLMs.** Adversarial attacks induce incorrect decisions in VLMs by applying elaborate and imperceptible perturbations to the input texts or images[18, 19, 20, 21, 22]. Text-based attacks [23, 24, 25, 26] mislead models into generating incorrect outputs through synonym

substitution, rewriting, or character-level perturbations. FGSM [27], PGD [28], AutoAttack [29], and C&W [30] are classical image-based white-box attacks that construct adversarial images by accessing model parameters and gradient information. In terms of multimodal attacks, Co-Attack [31] is a white-box attack method designed for VLMs, while more works focus on building transferable adversarial black-box attack frameworks [32, 33, 34, 35, 36, 37].

**General adversarial robustness.** Researchers have proposed multiple robustness strategies to enhance the reliability of models in adversarial settings [38, 39]. Detector-based approaches [40, 41] defend against adversarial attacks by detecting and filtering anomalous patterns within input samples. Purification methods [42, 43, 44] utilize techniques such as image transformations [45, 46] and denoising filters [47] to disrupt or remove potential adversarial perturbations from the input, yet they run the risk of weakening normal sample characteristics. Certified robustness approaches [48, 49, 50] provide theoretical and verifiable guarantees for model robustness, though they are typically applicable only to simple threat models with small certified radii. Adversarial training [51, 52, 53, 54] addresses model vulnerabilities by mining potential adversarial examples in the dataset and adapting the model to withstand adversarial attacks during the training process.

**Adversarial robustness of VLMs.** Numerous studies have explored the robustness of VLMs under adversarial attacks, mainly including defense strategies based on model fine-tuning and adversarial prompt tuning. TeCoA [5] and LAAT [6] enhance zero-shot adversarial robustness by leveraging the semantic consistency of the text encoder to guide fine-tuning of the image encoder. PMG-AFT [55] and FARE [56] leverage the generalization features of the original pre-trained model to improve the adversarial robustness of the CLIP visual encoder on downstream tasks while preserving natural generalizability. Prompt tuning serves as a lightweight adaptation approach that facilitates the efficient transfer of pretrained models toward the target task distribution [57, 15, 58, 59]. Recent studies [7, 8, 9] have shown that adversarial prompt tuning can efficiently enhance the robust adaptability of VLMs. APT [10] and AdvPT [11] approaches improve model robustness by introducing learnable textual prompts into the language branch of CLIP to align with adversarial image embeddings. Correspondingly, C-AVP [12] and TeCoA [5] incorporate learnable visual prompts to defend against adversarial attacks. Recent multimodal adversarial prompt methods [7, 13, 14, 8] enhance the consistency between visual and language features of adversarial examples under the guidance of pre-trained CLIP, thereby balancing natural generalization and robust adaptation.

## 3 Proposed Method

Although prompt learning preserves the general representations of pre-trained VLMs, the adapted prompts lead to overfitting on specific supervised tasks. We propose architectural refinements to enhance VLMs for achieving robustness in both in-distribution and OOD scenarios. Figure 1 provides an overview of our proposed approach, with further details presented in the following sections.

### 3.1 Preliminaries

**CLIP recap.** Let $\mathcal{V}_{\theta_v}(\cdot)$ and $\mathcal{T}_{\theta_t}(\cdot)$ denote the image encoder and text encoder of CLIP, respectively, where $\theta_v$ and $\theta_t$ represent the corresponding pre-trained weights. Given a natural image $v$, the input sequence for the visual branch is constructed as $\tilde{v} = \{v_{\mathrm{cls}}, v_{1:M}\}$, where $v_{1:M}$ are the patch-level linearly projections of the image, and $v_{\mathrm{cls}}$ is a learnable vector aggregating global features. Given a manually designed fixed text template $t$, the input sequence for the language branch is constructed as $\tilde{t} = \{t_{\mathrm{sos}}, t_{1:N}, t_c, t_{\mathrm{eos}}\}$, where $t_{1:N}$ and $t_c$ represent the word embeddings of the template text and the class label, respectively. $t_{\mathrm{sos}}$ and $t_{\mathrm{eos}}$ are non-parametric start and end tokens. The input sequences from the visual and language branches are encoded by CLIP in the latent space into image embeddings $\mathcal{V}_{\theta_v}(\tilde{v})$ and text embeddings $\mathcal{T}_{\theta_t}(\tilde{t})$, respectively. During zero-shot inference, the similarity between $\mathcal{V}_{\theta_v}(\tilde{v})$ and the text embeddings of all candidate categories $\{\mathcal{T}_{\theta_t}(\tilde{t}_c)\}_{c=1}^C$ is computed as $\frac{\exp\big(\mathrm{sim}\big(\mathcal{V}_{\theta_v}(\tilde{v}), \mathcal{T}_{\theta_t}(\tilde{t})\big)/\vartheta\big)}{\sum_{c=1}^C \exp\big(\mathrm{sim}\big(\mathcal{V}_{\theta_v}(\tilde{v}), \mathcal{T}_{\theta_t}(\tilde{t}_c)\big)/\vartheta\big)}$, where $\mathrm{sim}(\cdot, \cdot)$ denotes the cosine similarity function, $\vartheta$ is the temperature parameter, and $C$ is the total number of classes.

**Adversarial attacks against CLIP.** Given a natural image $v$ with ground-truth label $y$, adversaries construct a perceptually imperceptible adversarial example $v_{\mathrm{adv}} = v + \delta$ by optimizing the perturbation

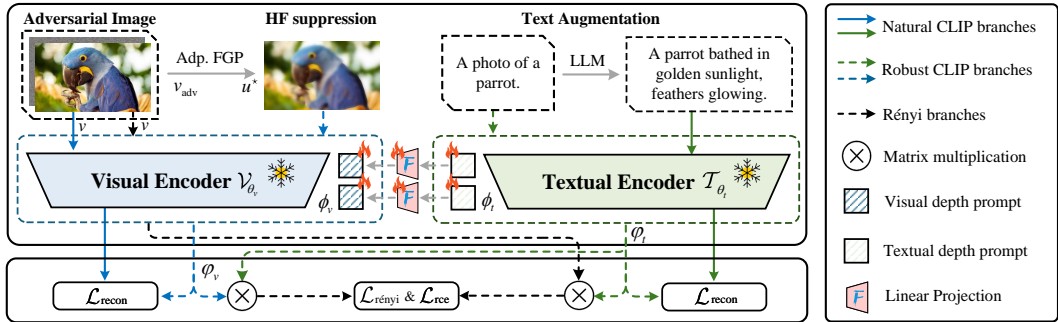

Figure 1: An overview of CoAPT. Natural CLIP processes natural images and extended descriptive text inputs. Robust CLIP takes as input the images subjected to HF suppression via the real-time Adaptive-FGP algorithm and restores the corrupted natural generalization features under the guidance of Natural CLIP in the latent space. The outputs of Robust CLIP are collaboratively regulated by the frozen CLIP weights $\theta$, the trainable deep multimodal adversarial prompts $\phi$, and the low-rank residual modules $\varphi$. The Rényi branch explicitly regulates the discrepancy between natural and adversarial distributions by calculating the divergence between their similarity scores.

$\delta$ within a $q$-norm ball of radius $\epsilon$. A successful attack must satisfy the following criteria:

$$\arg \max_{c \in \{1, \dots, C\}} \text{sim}(\mathcal{V}_{\theta_v}(\tilde{v}_{\text{adv}}), \mathcal{T}_{\theta_t}(\tilde{t}_c)) \neq y, \quad \text{s.t.} \quad \|v_{\text{adv}} - v\|_q \leq \epsilon. \tag{1}$$

**Adversarial prompt tuning.** APT enhances the adversarial adaptability of pre-trained VLMs for specific or novel downstream tasks by optimizing visual or textual prompts through adversarial training. Given the prompts $\phi = \{\phi_v^{1:V}, \phi_t^{1:T}\}$ to be optimized during adversarial training, where $V$ and $T$ represent the number of trainable tokens within the visual and textual prompts, respectively. Adversarial visual-only and text-only prompting [10, 11, 12] typically employs shallow prompting, where prompts are inserted solely into the input sequences. Specifically, the visual and textual input sequences are updated as $\tilde{v} = \{v_{\text{cls}}, \phi_v^{1:V}, v_{1:M}\}$ and $\tilde{t} = \{t_{\text{sos}}, \phi_t^{1:T}, t_c, t_{\text{eos}}\}$. Building upon shallow prompting, both independent and joint vision-language adversarial prompting [7, 8] incorporate deep prompts into multiple layers within the visual and language transformer architectures.

We aim to develop joint vision-language adversarial prompts that learn adversarial transformation-invariant features during training, strengthening the adversarial robustness of the CLIP visual branch. We still denote the adversarial deep prompts as $\phi$. Given a downstream dataset $\mathcal{D}$, $\phi$ is optimized jointly with the frozen parameters $\theta$ on adversarial examples. Focusing on the $\ell_\infty$ threat model, the adversarial optimization process for obtaining the optimal parameters of robust prompts $\phi^*$ can be formalized as:

$$\phi^* = \arg \min_\phi \mathbb{E}_{(v,y) \sim \mathcal{D}} \left[ \max_{\|v_{\text{adv}} - v\|_\infty \leq \epsilon} \mathcal{L}(\mathcal{V}_{\theta_v, \phi_v}(\tilde{v}_{\text{adv}}), \mathcal{T}_{\theta_t, \phi_t}(\tilde{t}_c)) \right]. \tag{2}$$

### 3.2 Real-Time Total Variation Regularization for High-Frequency Suppression

**Background on total variation.** Total variation regularization is implemented in the continuous and discrete settings by solving an unconstrained convex optimization problem in its penalized form:

$$\min_{u \in U} \frac{1}{2\lambda} \|u - v_{(\text{adv})}\|^2 + \|u\|_{\text{TV}}, \tag{3}$$

where $u \in U = \mathbb{R}^{m \times n}$ denotes the image to be restored, $v_{(\text{adv})} \in U$ represents either a natural or adversarial image. For simplicity, $v$ is used uniformly in this section. $\|\cdot\|_{\text{TV}}$ represents the discrete total variation of the image gradient, and $\lambda > 0$ balances the fidelity and regularization terms. Chambolle[60] transforms Eq. (3) into a nonlinear projection problem on a constrained space via dual formulation. However, this method lacks real-time capability and is prone to over-smoothing image details and residual adversarial perturbations. We design adaptive-FGP, a fast gradient projection (FGP) method with an adaptive restart mechanism and a spatially adaptive regularization strategy.

**Accelerated gradient method with adaptive restart mechanism.** We obtain the optimal solution from a norm-constrained dual vector field, thereby recovering $v$ in the form:

$$\min_{\mathbf{w} \in \mathcal{W}} \left\{ f(\mathbf{w}^k) := \left\| v - \gamma(v) \cdot \text{div}(\mathbf{w}^k) \right\|^2 \right\}, \tag{4}$$

where $k$ denotes the current time step, and $\mathcal{W} \subseteq \mathbb{R}^{(m-1) \times n} \times \mathbb{R}^{m \times (n-1)}$ is the unit-ball constraint set for the gradient dual components $\mathbf{w}_{i,j}^k = (p_{i,j}^{k,x}, \ p_{i,j}^{k,y})^\top$. If the gradient vector is defined in both horizontal and vertical directions, it satisfies $\|\mathbf{w}_{i,j}^k\| \leq 1$. Otherwise, only the single-direction constraint remains, satisfying $\|p_{i,n}^{k,x}\|_\infty \leq 1$ and $\|p_{m,j}^{k,y}\|_\infty \leq 1$. $\text{div}(\cdot)$ denotes the discrete divergence operator, which maps the dual variables $\mathbf{w}$ from the vector field $\mathcal{W}$ to the image domain $U$. The gradient of $f(\mathbf{w}^k)$ can be computed as $\nabla_{\mathbf{w}^k} f(\mathbf{w}^k) = -2 \cdot \gamma(v) \cdot \text{div}^* \left( v - \gamma(v) \cdot \text{div}(\mathbf{w}^k) \right)$. Using a step size of $1/L$, where $L$ denotes the Lipschitz constant of $f(\mathbf{w}^k)$ with its upper bound derived as $16\gamma^2(v)$ in the Appendix B. The dual variable update rule can be expressed as:

$$\mathbf{w}^k = \Pi_{\mathcal{W}} \left( \bar{\mathbf{w}}^k - \frac{\nabla(v - \gamma(v) \cdot \text{div}(\bar{\mathbf{w}}^k))}{8 \cdot \gamma(v)} \right), \tag{5}$$

where $\Pi_{\mathcal{W}}$ represents the projection operator. The update of $\bar{\mathbf{w}}$ is performed as follows:

$$\bar{\mathbf{w}}^{k+1} = \begin{cases} \mathbf{w}^k + (\tau_k - 1) \cdot \left( \mathbf{w}^k - \mathbf{w}^{k-1} \right) / \tau_{k+1}, & \text{if } \theta^k < \theta_{\text{th}}, \\ \mathbf{w}^k, & \text{otherwise}, \end{cases} \tag{6}$$

when $\theta^k$ meets the predefined threshold $\theta_{\text{th}}$, the Nesterov [61] time-scale variable is updated with $\tau_{k+1} = \left( 1 + \sqrt{1 + 4\tau_k^2} \right)/2$; otherwise, it is reset to 1.0. The solution to the objective function is denoted as $u^k = v - \gamma(v) \cdot \text{div}(\bar{\mathbf{w}}^{k-1})$. The solution increments at two consecutive time steps are defined as $\sigma_k = u^k - u^{k-1}$ and $\sigma_{k-1} = u^{k-1} - u^{k-2}$. Whether the current momentum accumulation benefits the variable update is determined utilizing a cosine similarity-based adaptive restart criterion:

$$\cos(\theta_k) = \frac{\langle \sigma_k, \sigma_{k-1} \rangle}{\|\sigma_k\| \cdot \|\sigma_{k-1}\| + \zeta}, \tag{7}$$

where $\zeta$ is a numerical stabilization term. When the angle between directions exceeds $90°$, it signals a sharp deviation or reversal between momentum and update, indicating trajectory discontinuity. We then reset the temporal scaling and disable momentum to avoid overshooting.

**Spatially adaptive regularization strategy.** The regularization map $\gamma(v) \in \mathbb{R}_+^{m \times n}$ is given by:

$$\gamma(v) = \mu_{\text{base}} \cdot \left( 1 + \mu_{\text{gain}} \cdot \Phi(v) \right), \tag{8}$$

where $\mu_{\text{base}}, \mu_{\text{gain}} \in \mathbb{R}^+$ represent the base regularization strength and the sensitivity of the adjustment factor, respectively. The edge magnitude response function $\Phi(v) \in \mathbb{R}_+^{m \times n}$ is estimated using Sobel convolution kernels as $\sqrt{(v * K_x)^2 + (v * K_y)^2}$, where $K_x$ and $K_y$ denote the horizontal and vertical Sobel operators respectively. This adaptive regularization strategy automatically reduces the regularization strength in edge regions while enhancing it in flat regions, thereby preserving structural image details and effectively suppressing adversarial perturbations.

**Convergence criterion.** The relative change in update is measured through the Frobenius norm:

$$\max_{i \in \{k, k-1, ..., k-s\}} \frac{\|\sigma_i\|_F}{\|u^i\|_F + \zeta} < \xi. \tag{9}$$

If the convergence tolerance threshold $\xi > 0$ is satisfied for $s$ consecutive iterations, the projection optimization problem is considered to have converged. Based on the optimal solution $\mathbf{w}^{k\star}(v)$, the optimal image estimate for the original problem can be recovered as $\rho(v) = v - \gamma(v) \cdot \text{div}(\mathbf{w}^{k\star}(v))$.

### 3.3 Natural-Latent-Guided Adversarial Prompt Learning

**Reconstruction of natural generalization representations.** CoAPT employs deep contextual multimodal prompts and refines visual prompts through linear projection onto language prompts

to foster synergy between visual-language prompts. As illustrated in Figure 1, we efficiently learn generalizable knowledge from the natural CLIP by aligning its clean vision-language embeddings with adversarial embeddings from the robust CLIP in the latent space. Notably, Vanilla CLIP employs fixed text templates, which limit its ability to capture the semantic diversity required for generalization effectively during fine-tuning. A Gaussian radial basis function (RBF) is used to measure the embedding similarity between the natural CLIP and the robust CLIP in the latent space. Compared to cosine similarity, which primarily captures angular differences of vectors, Gaussian RBF highlights feature shifts caused by small-scale perturbations, allowing more sensitive detection of subtle distributional changes. In particular, we align both the visual and language branches:

$$\mathcal{L}_{\text{recon}} = 2 - \exp\left(-\beta\left(\|\mathcal{V}_{\theta_v,\phi_v,\varphi_v}(\rho(\tilde{v}_{\text{adv}})) - \mathcal{V}_{\theta_v}(\tilde{v})\|_2^2 + \|\mathcal{T}_{\theta_t,\phi_t,\varphi_t}(\tilde{t}) - \mathcal{T}_{\theta_t}(\tilde{t})\|_2^2\right)\right), \quad (10)$$

where the parameter $\beta = (2\sigma^2)^{-1}$ controls the sensitivity of distance variation to similarity. $\varphi_v$ and $\varphi_t$ are the low-rank residual modules introduced next. The learnable prompts in both the language and visual branches can adapt the data distribution of Vanilla CLIP to that of specific downstream adversarial tasks, while preserving and enhancing generalization and robustness to OOD tasks.

**Low-rank residual module.**  Directly imposing consistency constraints in the latent space is equivalent to introducing a strong supervisory signal, which lacks the flexibility to adapt to task-specific requirements and interpretable deviations. Inspired by LoRA [62], we introduce two low-rank matrices as an intermediate learnable bottleneck structure. This design allows the model to preserve the backbone features while selectively capturing fine-grained task-specific shifts within a compact subspace. Specifically, we incorporate an additional update term through low-rank reparameterization:

$$\mathcal{V}_{\theta,\phi,\varphi} = (I + \eta \cdot BA)\mathcal{V}_{\theta,\phi}, \quad (11)$$

where $\eta$ is the scaling factor, $B \in \mathbb{R}^{d \times r}$, $A \in \mathbb{R}^{r \times d}$, and $r \ll d$. The initial parameter perturbation is controlled by initializing the matrices as $A \sim \mathcal{N}(0, 1/r)$ and $B \sim \delta(0)$.

**Rényi regularization.**  Let $P$ and $Q$ denote the predicted probability distributions of natural and adversarial samples in the vision-language space of robust CLIP, respectively. Since adversarial samples are derived from minor perturbations of natural samples, $P$ is considered absolutely continuous with respect to $Q$. We introduce a regularization loss based on the $\alpha$-order Rényi divergence [63] to reduce the discrepancy between the natural and adversarial predictive distributions in robust CLIP:

$$\mathcal{L}_{\text{rényi}} = \frac{1}{\alpha - 1} \log \mathbb{E}_P\left[\left(\frac{dP}{dQ}\right)^{\alpha-1}\right], \alpha \in [0, \infty), \quad (12)$$

where $\frac{dP}{dQ}$ is the Radon–Nikodym derivative of $P$ with respect to $Q$. $\alpha$ explicitly controls the sensitivity to distributional differences. Higher orders ($\alpha > 1$) enhance the ability of the model to suppress spurious correlations. This mechanism corrects potential discriminative boundary ambiguities and reduces overfitting risks by preserving task-beneficial generalized features. Correspondingly, the supervised loss for downstream classification tasks can be expressed with the Rényi cross-entropy [64]:

$$\mathcal{L}_{\text{rce}} = \frac{\alpha}{1 - \alpha} \log \sum_i P(i) \cdot Q(i)^{\frac{\alpha-1}{\alpha}}, \quad \alpha \in [0, \infty). \quad (13)$$

Note that the Rényi cross entropy degenerates into Shannon cross entropy when the dataset labels are represented in one-hot coding. The overall training objective of CoAPT can be expressed as follows:

$$\mathcal{L}_{\text{coapt}} = \kappa_1 \mathcal{L}_{\text{recon}} + \kappa_2 \mathcal{L}_{\text{rényi}} + \kappa_3 \mathcal{L}_{\text{rce}}, \quad (14)$$

$\kappa_1, \kappa_2, \kappa_3$ are hyperparameters weighting contributions of individual losses to the overall objective.

**Overview of proposed method.**  Algorithm 1 illustrates the adversarial prompt optimization procedure adopted by CoAPT. In each training iteration, a batch of image-label pairs $(v, y)$ is sampled from the downstream dataset $\mathcal{D}$. Subsequently, the visual and textual sequences are constructed and accompanied by trainable deep prompts. Perceptually invisible adversarial examples $v_{\text{adv}}$ are crafted under $\ell_\infty$ norm constraints to induce erroneous model predictions (Lines 2∼4). These sequences are then fed into the natural CLIP and the robust CLIP equipped with low-rank residual modules $\varphi_v$ and $\varphi_t$ to obtain the corresponding visual and language representations (Lines 5∼9). CoAPT integrates

---

**Algorithm 1** Natural-Latent-Guided Adversarial Prompt Learning

---

**Input:** Dataset $\mathcal{D}$, frozen CLIP encoders $\mathcal{V}_{\theta_v}, \mathcal{T}_{\theta_t}$, prompt parameters $\phi = \{\phi_v, \phi_t\}$, low-rank
    modules $\varphi = \{\varphi_v, \varphi_t\}$, loss weights $\kappa_1, \kappa_2, \kappa_3$, adversarial budget $\epsilon$
**Output:** Optimized robust prompts $\phi^\star$
 1: **for** each minibatch $(v, y) \sim \mathcal{D}$ **do**
 2:    Set the real-time total variation regularization parameters
 3:    Construct input sequences $\tilde{v}, \tilde{t}$ and deep prompts $\phi$
 4:    Generate adversarial example $v_{\text{adv}}$ under $\ell_\infty$ constraint: $\|v_{\text{adv}} - v\|_\infty \leq \epsilon$
 5:    Generate visual and textual representations for natural CLIP and robust CLIP:
 6:       $\mathcal{V}_{\text{nat}} \leftarrow \mathcal{V}_{\theta_v}(\tilde{v})$
 7:       $\mathcal{T}_{\text{nat}} \leftarrow \mathcal{T}_{\theta_t}(\tilde{t})$
 8:       $\mathcal{V}_{\text{adv}} \leftarrow \mathcal{V}_{\theta_v, \phi_v, \varphi_v}(\rho(\tilde{v}_{\text{adv}}))$
 9:       $\mathcal{T}_{\text{adv}} \leftarrow \mathcal{T}_{\theta_t, \phi_t, \varphi_t}(\tilde{t})$
10:    Compute reconstruction loss $\mathcal{L}_{\text{recon}} \leftarrow 2 - \exp\left(-\beta(\|\mathcal{V}_{\text{adv}} - \mathcal{V}_{\text{nat}}\|_2^2 + \|\mathcal{T}_{\text{adv}} - \mathcal{T}_{\text{nat}}\|_2^2)\right)$
11:    Compute visual-textual representation similarity $P = scale \cdot \mathcal{V}_{\text{nat}} \cdot \mathcal{T}_{\text{nat}}^\top, Q = scale \cdot \mathcal{V}_{\text{adv}} \cdot \mathcal{T}_{\text{adv}}^\top$
12:    Compute Rényi divergence loss $\mathcal{L}_{\text{rényi}} \leftarrow \frac{1}{\alpha-1} \log \mathbb{E}_P[(\frac{dP}{dQ})^{\alpha-1}]$
13:    Compute Rényi cross-entropy loss: $\mathcal{L}_{\text{rce}} \leftarrow \frac{\alpha}{1-\alpha} \log \sum_i P(i) \cdot Q(i)^{\frac{\alpha-1}{\alpha}}$
14:    Take gradient step on $\nabla_{\phi, \varphi}(\kappa_1 \mathcal{L}_{\text{recon}} + \kappa_2 \mathcal{L}_{\text{rényi}} + \kappa_3 \mathcal{L}_{\text{rce}})$
15:    $\phi, \varphi \leftarrow \text{Backward}(\nabla_{\phi, \varphi})$
16: **end for**

---

three losses, including a reconstruction loss for recovering generalization, a Rényi divergence loss to quantify prediction discrepancies between natural and adversarial samples, and a cross-entropy loss for classification (Lines 10∼13). Finally, only the prompt parameters $\phi$ and the low-rank module parameters $\varphi$ are updated via gradient descent. Adversarial prompt learning significantly improves the robust generalization of the model under image perturbations and distributional shifts, and exhibits strong cross-task transferability (Lines 14∼15).

## 4 Experiments

### 4.1 Evaluation Settings

**Datasets and benchmark settings.** We conduct a comprehensive evaluation of the proposed CoAPT method across four benchmark settings on 15 datasets spanning diverse vision tasks. For the evaluation of few-shot learning, base-to-novel class generalization, and zero-shot benchmarks, we adopt 11 image classification datasets, including EuroSAT [65] for satellite imagery, UCF101 [66] for action recognition, DTD [67] for texture classification, SUN397 [68] for scene recognition, Caltech101 [69] and ImageNet [70] for general object recognition, and FGVC Aircraft [71], Flowers102 [72], OxfordPets [73], Food101 [74], and StanfordCars [75] for fine-grained classification tasks. For the OOD benchmark, we select four variants of ImageNet, ImageNet-A [76], ImageNet-R [77], ImageNet-Sketch [78], and ImageNetV2 [79], as the domain generalization test sets. Notably, both zero-shot and OOD utilize the training set of ImageNet as the source dataset.

**Adversarial training and evaluation.** The attack settings of baseline methods TeCoA [5] and FAP [7] are adopted to ensure fair comparison. During adversarial training, we adopt a two-step PGD attack with a maximum perturbation magnitude $\ell_\infty = 1/255$ and step size $\alpha = 1/255$. For robustness evaluation, we employ a 100-step PGD attack under the same constraints to thoroughly assess the defense capability of the model under strong attacks.

**Implementation details.** Our method is built upon the ViT-B/32 architecture of Vanilla CLIP. Each experiment is conducted three times with different random seeds, and the average results are reported. The convergence tolerance threshold in Adaptive-FGP is set to $\xi = 1e^{-3}, s = 3$, and the maximum number of iterations is 30. The parameters of the regularization factor map $\gamma(v)$ are set to $\mu_{\text{base}} = 0.1$ and $\mu_{\text{gain}} = 1.2$. We employed 2.5-order Rényi divergence regularization, with $\mathcal{L}_{\text{coapt}}$ coefficients set to $\kappa_1 = 8, \kappa_2 = 1, \kappa_3 = 1$. Adversarial prompts with a length of 4 and a depth of 9 are applied to both the visual and textual branches. The RAdam optimizer with an initial learning rate of 0.00735

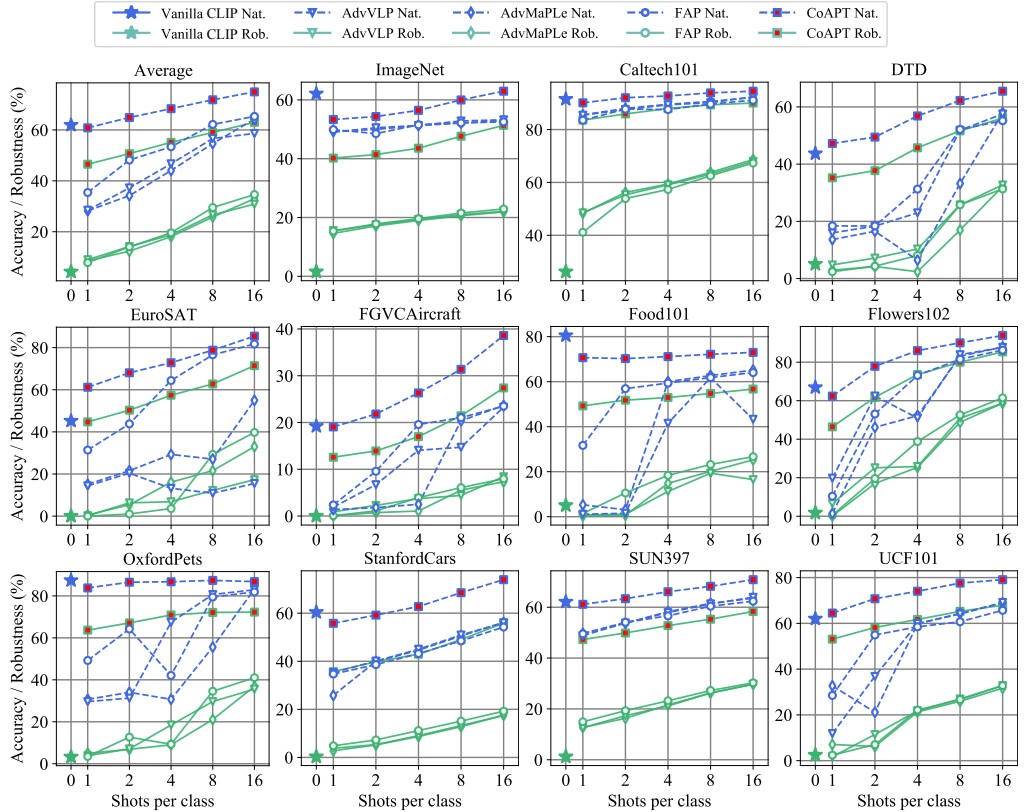

Figure 2: The few-shot performance across 11 benchmark datasets under varying numbers of shots.

is adopted, and the batch size is set to 64. In contrast to the existing research work, we do not set proprietary hyperparameters for any of the benchmarks and datasets, in order to prove the generality of the proposed CoAPT. Under few-shot settings we compare with FAP and baselines from its paper.

## 4.2 Adversarial Few-Shot Learning

The robust generalization capability of each model to specific tasks is evaluated under the condition of only a few identically distributed samples. As shown in Figure 2, CoAPT demonstrates consistently superior performance compared to all baseline methods. CoAPT exhibits robust learning ability with near-linear steady improvement in natural and adversarial accuracy as the number of shots increases. In contrast, the baseline methods show significant performance fluctuations across different shot counts. Furthermore, our approach achieves superior control over the trade-off between natural accuracy and adversarial robustness. In most of the datasets, CoAPT is able to match the natural accuracy of Vanilla CLIP with only 1-shot learning. On six datasets, including Caltech101, our robust accuracy is even higher than the natural accuracy of the baseline method. The robust accuracy of CoAPT on five datasets, including DTD, can be improved to higher than the natural accuracy of Vanilla CLIP by few-shot learning.

## 4.3 Adversarial Base-to-New Generalization

We assess the ability of the models to balance robust adaptation to specific class distributions and robust generalization to unseen class distributions. Specifically, the models are trained on base classes with a 16-shot setting and jointly evaluated on the base classes and the novel unseen classes. As shown in Table 1, our method outperforms state-of-the-art approaches on all datasets. While improving the average harmonic mean (HM) of robustness by 32.39%, the natural generalization performance of the model also achieves an average gain of 13.09%. Notably, the harmonic mean of robustness for novel classes reaches a maximum of 51.57% on the OxfordPets dataset. These

Table 1: Comparison with state-of-the-art methods on base-to-novel generalization. *Gain* denotes the absolute performance improvement.

**(a) Average**

| | | Acc. | FAP | CoAPT | Gain↑ |
|---|---|---|---|---|---|
| Nat. | Base | 70.52 | 78.47 | | **7.95** |
| | Novel | 49.58 | 65.35 | | **15.77** |
| | HM | 58.22 | 71.31 | | **13.09** |
| Rob. | Base | 38.05 | 67.70 | | **29.65** |
| | Novel | 21.86 | 54.13 | | **32.27** |
| | HM | 27.77 | 60.16 | | **32.39** |

**(b) ImageNet**

| | | Acc. | FAP | CoAPT | Gain↑ |
|---|---|---|---|---|---|
| Nat. | Base | 58.10 | 66.15 | | **8.05** |
| | Novel | 47.83 | 55.41 | | **7.58** |
| | HM | 52.47 | 60.30 | | **7.84** |
| Rob. | Base | 25.83 | 52.65 | | **26.82** |
| | Novel | 21.57 | 45.07 | | **23.50** |
| | HM | 23.51 | 48.57 | | **25.06** |

**(c) Caltech101**

| | | Acc. | FAP | CoAPT | Gain↑ |
|---|---|---|---|---|---|
| Nat. | Base | 94.07 | 97.25 | | **3.18** |
| | Novel | 76.53 | 92.72 | | **16.19** |
| | HM | 84.40 | 94.93 | | **10.53** |
| Rob. | Base | 74.20 | 94.38 | | **20.18** |
| | Novel | 50.00 | 88.03 | | **38.03** |
| | HM | 59.74 | 91.09 | | **31.35** |

**(d) DTD**

| | | Acc. | FAP | CoAPT | Gain↑ |
|---|---|---|---|---|---|
| Nat. | Base | 69.17 | 76.08 | | **6.91** |
| | Novel | 35.17 | 54.03 | | **18.86** |
| | HM | 46.63 | 63.17 | | **16.54** |
| Rob. | Base | 41.63 | 67.98 | | **26.35** |
| | Novel | 19.77 | 43.88 | | **24.11** |
| | HM | 26.81 | 53.31 | | **26.50** |

**(e) EuroSAT**

| | | Acc. | FAP | CoAPT | Gain↑ |
|---|---|---|---|---|---|
| Nat. | Base | 87.70 | 91.61 | | **3.91** |
| | Novel | 32.80 | 56.11 | | **23.31** |
| | HM | 47.74 | 69.33 | | **21.59** |
| Rob. | Base | 51.80 | 84.67 | | **32.87** |
| | Novel | 13.40 | 47.40 | | **34.00** |
| | HM | 21.29 | 60.55 | | **39.25** |

**(f) FGVCAircraft**

| | | Acc. | FAP | CoAPT | Gain↑ |
|---|---|---|---|---|---|
| Nat. | Base | 24.83 | 35.37 | | **10.54** |
| | Novel | 15.83 | 25.41 | | **9.58** |
| | HM | 19.33 | 29.58 | | **10.24** |
| Rob. | Base | 8.00 | 25.37 | | **17.37** |
| | Novel | 4.23 | 16.68 | | **12.45** |
| | HM | 5.53 | 20.12 | | **14.59** |

**(g) Food101**

| | | Acc. | FAP | CoAPT | Gain↑ |
|---|---|---|---|---|---|
| Nat. | Base | 72.37 | 78.20 | | **5.83** |
| | Novel | 68.20 | 79.47 | | **11.27** |
| | HM | 70.22 | 78.83 | | **8.60** |
| Rob. | Base | 27.57 | 62.03 | | **34.46** |
| | Novel | 24.20 | 62.86 | | **38.66** |
| | HM | 25.78 | 62.44 | | **36.66** |

**(h) Flowers102**

| | | Acc. | FAP | CoAPT | Gain↑ |
|---|---|---|---|---|---|
| Nat. | Base | 89.30 | 94.94 | | **5.64** |
| | Novel | 45.67 | 63.07 | | **17.40** |
| | HM | 60.43 | 75.79 | | **15.36** |
| Rob. | Base | 65.50 | 88.57 | | **23.07** |
| | Novel | 18.10 | 51.89 | | **33.79** |
| | HM | 28.36 | 65.42 | | **37.06** |

**(i) OxfordPets**

| | | Acc. | FAP | CoAPT | Gain↑ |
|---|---|---|---|---|---|
| Nat. | Base | 87.37 | 90.55 | | **3.18** |
| | Novel | 72.13 | 94.50 | | **22.37** |
| | HM | 79.02 | 92.48 | | **13.46** |
| Rob. | Base | 34.13 | 78.72 | | **44.59** |
| | Novel | 26.07 | 83.71 | | **57.64** |
| | HM | 29.56 | 81.13 | | **51.57** |

**(j) StanfordCars**

| | | Acc. | FAP | CoAPT | Gain↑ |
|---|---|---|---|---|---|
| Nat. | Base | 53.97 | 73.34 | | **19.37** |
| | Novel | 42.67 | 59.20 | | **16.53** |
| | HM | 47.66 | 65.51 | | **17.85** |
| Rob. | Base | 18.60 | 54.20 | | **35.60** |
| | Novel | 14.10 | 40.95 | | **26.85** |
| | HM | 16.04 | 46.65 | | **30.61** |

**(k) SUN397**

| | | Acc. | FAP | CoAPT | Gain↑ |
|---|---|---|---|---|---|
| Nat. | Base | 68.47 | 76.69 | | **8.22** |
| | Novel | 61.47 | 70.46 | | **8.99** |
| | HM | 64.78 | 73.44 | | **8.66** |
| Rob. | Base | 34.63 | 64.50 | | **29.87** |
| | Novel | 30.77 | 58.50 | | **27.73** |
| | HM | 32.59 | 61.35 | | **28.76** |

**(l) UCF101**

| | | Acc. | FAP | CoAPT | Gain↑ |
|---|---|---|---|---|---|
| Nat. | Base | 70.37 | 82.95 | | **12.58** |
| | Novel | 47.10 | 68.45 | | **21.35** |
| | HM | 56.43 | 75.00 | | **18.57** |
| Rob. | Base | 36.63 | 71.65 | | **35.02** |
| | Novel | 18.30 | 56.50 | | **38.20** |
| | HM | 24.41 | 63.18 | | **38.77** |

results demonstrate that the robust prompts learned by CoAPT not only adapt to category-specific distributional shifts and distributional discrepancies between natural and adversarial examples but also effectively preserve the natural generalization capability of the original pretrained model.

## 4.4 Zero-Shot Performance

The generalization ability of the models across datasets is explored. CoAPT is trained on ImageNet as the source dataset and then evaluated on ten different types of downstream target datasets. The evaluation for each dataset and the corresponding statistical results are presented in Figure 3 and Table 2, respectively. Compared to the FAP method, our approach achieves significant improvements across all metrics on all datasets, particularly in adversarial robustness. We attain a better trade-off between natural and adversarial generalization. Relative to Vanilla CLIP, we sacrifice only 7.83% in natural generalization accuracy while achieving absolute gains of 49.61% and 39.37% in robustness on the source and target datasets.

Table 2: CoAPT performance on source dataset and average results across 10 target datasets.

| Method | ImageNet | | Average | |
|---|---|---|---|---|
| | Nat. | Rob. | Nat. | Rob. |
| CLIP | 62.10 | 1.57 | **61.89** | 4.53 |
| FAP | 50.80 | 21.60 | 45.72 | 23.89 |
| CoAPT | **63.42**$_{1.32↑}$ | **51.18**$_{29.58↑}$ | 54.06$_{7.83↓}$ | **43.90**$_{20.01↑}$ |

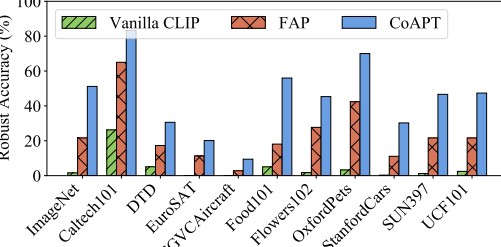
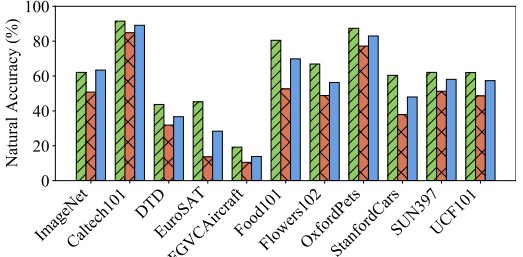

Figure 3: Zero-shot robust and natural accuracies on the source and 10 target datasets.

## 4.5 Out-of-Distribution Performance

We test the natural generalization and adversarial robustness of the model under domain distribution shift. While maintaining ImageNet as the source dataset, we conduct direct evaluations on four representative variant datasets that share the same set of categories. As shown in Table 3, our method achieves superior natural generalization and robust adaptation across all target datasets compared to the comparison methods.

Table 3: Comparison of OOD generalization performance.

| Method | ImageNet-A | | ImageNet-R | | ImageNet-Sketch | | ImageNet-V2 | | Average | |
|---|---|---|---|---|---|---|---|---|---|---|
| | Nat. | Rob. | Nat. | Rob. | Nat. | Rob. | Nat. | Rob. | Nat. | Rob. |
| FAP | 9.40 | 1.20 | 51.60 | 28.20 | 28.40 | 16.30 | 42.80 | 16.60 | 33.05 | 15.575 |
| CoAPT | $\mathbf{16.99}_{7.59\uparrow}$ | $\mathbf{9.72}_{8.52\uparrow}$ | $\mathbf{60.35}_{8.75\uparrow}$ | $\mathbf{50.71}_{22.51\uparrow}$ | $\mathbf{35.76}_{7.36\uparrow}$ | $\mathbf{29.13}_{12.83\uparrow}$ | $\mathbf{54.35}_{11.55\uparrow}$ | $\mathbf{42.23}_{25.63\uparrow}$ | $\mathbf{41.86}_{8.81\uparrow}$ | $\mathbf{32.95}_{17.37\uparrow}$ |

## 4.6 Ablation Analysis

As shown in Table 4, we progressively ablate CoAPT components to evaluate their generalizability and importance across the four benchmarks. CoAPT with all components achieves the best performance on all benchmarks. We first remove the adaptive restart mechanism. Most metrics exhibited varying degrees of degradation, with 16-shot and OOD robust accuracy declining by 1.85% and 1.69%, respectively. This mechanism restores optimal convergence without prior knowledge of function parameters and enhances stability near the optimum. We replace the spatially adaptive regularization strategy with a fixed global regularization factor. The ablated model ignores the diversity of image spatial structures, leading to structural blurring and loss of details, with an average drop of 6.00% in clean accuracy across the four benchmarks. We subsequently remove the entire adaptive-FGP method, thereby eliminating the adversarial space compression. During high-level feature recovery in the natural CLIP latent space, the model places greater emphasis on high-frequency components where adversarial perturbations are concentrated, resulting in a degradation in adversarial robustness. However, even with full natural images, the ablated model yields lower natural accuracy than full CoAPT across all benchmarks. Removing the low-rank residual module leads to drops in few-shot-16 robustness and base-to-novel accuracy. As it is sensitive to dataset-specific hyperparameters and was not fine-tuned, its effectiveness is limited. However, due to its potential on certain datasets, the module is retained. When we remove Rényi regularization, the overall performance of the model decreases. Rényi regularization facilitates early detection and correction of boundary ambiguities, and mitigates overfitting by preserving task-relevant generalizable features. CoAPT reduces to a TeCoA-like approach when the final reconstruction loss is removed. The performance drop on unseen tasks is due to the reconstruction loss guiding prompts toward task-irrelevant generalization.

Table 4: Ablation study of CoAPT components on 15 datasets across four benchmarks.

| Ablation term | Few-shot-16 | | Base-to-novel | | | | | | Zero-shot | | OOD | |
|---|---|---|---|---|---|---|---|---|---|---|---|---|
| | Nat. | Rob. | Nat. | Rob. | HM | Nat. | Rob. | HM | Nat. | Rob. | Nat. | Rob. |
| No ablation | 74.96 | 62.98 | 78.47 | 67.70 | 72.69 | 65.35 | 54.13 | 59.21 | 54.91 | 44.57 | 41.86 | 32.95 |
| Adp. rst. | 74.82 | 61.13 | 78.31 | 66.73 | 72.06 | 65.56 | 53.40 | 58.86 | 54.38 | 43.06 | 40.91 | 31.26 |
| Adp. reg. | 68.86 | 63.34 | 73.33 | 68.03 | 70.58 | 57.74 | 52.87 | 55.20 | 49.16 | 44.15 | 34.86 | 31.70 |
| Adp. FGP | 74.34 | 31.64 | 78.15 | 35.92 | 49.21 | 64.36 | 24.41 | 35.40 | 53.00 | 18.65 | 38.52 | 13.21 |
| Res. mod. | 74.64 | 31.41 | 78.33 | 35.99 | 49.32 | 64.18 | 26.00 | 37.01 | 55.07 | 19.74 | 41.04 | 14.31 |
| Rényi | 73.09 | 30.73 | 78.18 | 33.66 | 47.06 | 63.57 | 24.97 | 35.86 | 55.30 | 19.84 | 41.05 | 14.55 |
| Recon. loss | 71.82 | 31.47 | 76.66 | 34.52 | 47.60 | 58.71 | 22.25 | 32.27 | 51.85 | 19.85 | 38.66 | 13.91 |

## 5 Conclusion

We focus on the adversarial robustness of VLMs and propose a novel adversarial prompt tuning paradigm in which pre-trained VLMs collaborate with target robust VLMs. CoAPT begins with a proposed real-time adaptive TV regularization algorithm to attenuate high-frequency details of the input images to compress the perturbation space of the adversarial samples. Subsequently, under the guidance of natural CLIP, CoAPT restores the natural generalization features disrupted by adversarial perturbations in the latent representation space. CoAPT achieves an effective trade-off among natural generalization, adversarial robustness, and task-specific adaptation. The overall performance of CoAPT significantly surpasses that of current state-of-the-art methods on 15 datasets across the benchmarks of few-shot, base-to-novel, zero-shot, and out-of-distribution generalization.

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

# Appendix

## A   Pipelines of Adaptive-FGP Algorithm

Algorithm 2 presents the proposed adaptive fast gradient projection (adaptive-FGP) method for real-time total variation regularization. It is designed to disrupt the perturbation space of adversarial examples while maximally preserving the structural integrity of image content.

---

**Algorithm 2** Real-Time Total Variation Regularization Based on Proposed Adaptive-FGP Algorithm

---

**Input:** Image $v_{(adv)}$, base coefficient $\mu_{\text{base}}$, gain $\mu_{\text{gain}}$, convergence tolerance $\xi$
**Output:** Recovered image $u^{\star}$
1: Compute $\gamma(v) = \mu_{\text{base}} \cdot (1 + \mu_{\text{gain}} \cdot \Phi(v))$ on image $v$ using Sobel operator
2: Initialize $\mathbf{w}^0 = \mathbf{0}$, $\bar{\mathbf{w}}^0 = \mathbf{0}$, $\tau_0 = 1$
3: **for** $k = 1$ to Maximum iterations **do**
4:     Compute $u^k = v - \gamma(v) \cdot \text{div}(\bar{\mathbf{w}}^{k-1})$
5:     Compute gradient $\nabla f(\bar{\mathbf{w}}^k) = -2 \cdot \gamma(v) \cdot \text{div}^*(u^k)$
6:     Update $\mathbf{w}^k = \Pi_{\mathcal{W}} \left( \bar{\mathbf{w}}^k - \nabla(u^k)/8 \cdot \gamma(v) \right)$
7:     Compute $\sigma_k = u^k - u^{k-1}$, $\sigma_{k-1} = u^{k-1} - u^{k-2}$
8:     Compute $\cos(\theta_k) = \langle \sigma_k, \sigma_{k-1} \rangle / (\|\sigma_k\| \cdot \|\sigma_{k-1}\| + \zeta)$
9:     **if** $\cos(\theta_k) > \cos(\theta_{\text{th}})$ **then**
10:         $\tau_{k+1} = (1 + \sqrt{1 + 4\tau_k^2})/2$
11:         $\mathbf{w}^k + (\tau_k - 1) \cdot \left( \mathbf{w}^k - \mathbf{w}^{k-1} \right) / \tau_{k+1}$
12:     **else**
13:         $\tau_{k+1} = 1$, $\bar{\mathbf{w}}^{k+1} = \mathbf{w}^k$
14:     **end if**
15:     **if** $\max_{i \in \{k, \ldots, k-s\}} \|\sigma_i\|_F / (\|u^i\|_F + \zeta) < \xi$ **then**
16:         **break**
17:     **end if**
18: **end for**
19: **return** $u^{\star} = u^k$

---

**Initialization phase.** The algorithm first constructs a spatially adaptive regularization map $\gamma(v)$ based on the input image $v$. This regularization term is governed by a baseline intensity coefficient $\mu_{\text{base}}$ and an edge sensitivity coefficient $\mu_{\text{gain}}$, with the edge response $\Phi(v)$ is estimated via the Sobel convolution operator. This strategy automatically reduces the regularization strength in edge regions to preserve structural details, while enhancing regularization intensity in the flat areas to effectively suppress adversarial perturbations. Subsequently, the dual variable $\mathbf{w}^0$ and its accelerated counterpart $\bar{\mathbf{w}}^0$, along with the temporal scaling factor $\tau_0$ are initialized (Lines 1~2).

**Gradient projection update in the dual space.** First, the dual variable field from the previous iteration is transformed into a scalar field via the divergence operator, which is utilized to construct the current estimate of the primal variable image $u^k$. Subsequently, the dual variable $\bar{\mathbf{w}}^k$ at the current iteration is updated and projected onto the dual constraint set $\mathcal{W}$ to ensure that the gradient field satisfies the unit ball constraint (Lines 4~6).

**Momentum acceleration with adaptive restart mechanism.** We measure whether the direction of the angle between two consecutive step increments $\sigma_k$ and $\sigma_{k-1}$ is reversed to determine whether a restart has occurred. If no deviation in direction is detected, the Nesterov momentum acceleration mechanism is applied to enhance convergence speed. Otherwise, if the angle between directions exceeds a predefined threshold, the momentum accumulation is reset to prevent overshooting caused by trajectory discontinuity, thereby improving the stability of the algorithm. The adaptive restart mechanism originates from an analysis of oscillatory behavior inherent in Nesterov-type momentum schemes, which is particularly important in the context of spatially weighted total variation with non-uniform regularization terms (Lines 7~14).

**Convergence criterion.** The algorithm is deemed to have converged when the relative change in updates, measured by the Frobenius norm, remains below the threshold $\xi$ for $s$ consecutive iterations. This condition ensures the stability of the solution across multiple time steps in the output image while effectively avoiding redundant iterations. Upon completion of the iterations, the optimal solution

$u^\star$ of the output image is obtained. The adaptive-FGP method exhibits strong parallelizability and efficient acceleration mechanisms, significantly enhancing model robustness while keeping the computational overhead below 10% (Lines 15~17).

## B  Upper Bound Analysis of the Lipschitz Constant

Since the gradient $\nabla f(\boldsymbol{w})$ of the objective function $f(\boldsymbol{w})$ is Lipschitz continuous, there exists a constant $L > 0$ such that for any $\boldsymbol{w}_1, \boldsymbol{w}_2$, the following inequality holds:

$$\|\nabla f(\boldsymbol{w}_1) - \nabla f(\boldsymbol{w}_2)\| \le L\|\boldsymbol{w}_1 - \boldsymbol{w}_2\|. \tag{15}$$

The gradient difference can be computed as:

$$
\begin{aligned}
\nabla f(\boldsymbol{w}_1) - \nabla f(\boldsymbol{w}_2) &= -2\gamma(v) \cdot \nabla \operatorname{div}^* \left[ (v - \gamma(v) \cdot \operatorname{div}(\boldsymbol{w}_1)) - (v - \gamma(v) \cdot \operatorname{div}(\boldsymbol{w}_2)) \right] \\
&= 2\gamma(v)^2 \cdot \nabla \operatorname{div}^* \left[ \operatorname{div}(\boldsymbol{w}_1) - \operatorname{div}(\boldsymbol{w}_2) \right] \\
&= 2\gamma(v)^2 \cdot \nabla \operatorname{div}^* \cdot \operatorname{div}(\boldsymbol{w}_1 - \boldsymbol{w}_2).
\end{aligned}
\tag{16}
$$

Thus, the norm is bounded by:

$$
\begin{aligned}
\|\nabla f(\boldsymbol{w}_1) - \nabla f(\boldsymbol{w}_2)\| &\le 2\gamma(v)^2 \cdot \|\nabla \operatorname{div}^T \cdot \operatorname{div}\| \cdot \|\boldsymbol{w}_1 - \boldsymbol{w}_2\| \\
&\le 2\gamma(v)^2 \cdot \|\operatorname{div}\|^2 \cdot \|\boldsymbol{w}_1 - \boldsymbol{w}_2\|.
\end{aligned}
\tag{17}
$$

Analogous to the spectral norm bound of the discrete gradient operator in the TV regularization term, if the operator norm of the discrete divergence operator satisfies $|\operatorname{div}| \le \sqrt{8}$, we obtain:

$$\|\nabla f(\boldsymbol{w}_1) - \nabla f(\boldsymbol{w}_2)\| \le 16\gamma(v)^2 \cdot \|\boldsymbol{w}_1 - \boldsymbol{w}_2\|. \tag{18}$$

Therefore, the upper bound of the Lipschitz constant $L(f)$ for the objective function $f(\boldsymbol{w})$ is given by:

$$L(f) \le 16\gamma(v)^2. \tag{19}$$

## C  Additional Experimental Results

### C.1  Sensitivity Analysis of PGD Attack Hyperparameters

Table 5 systematically evaluates the impact of different configurations on natural and robust accuracy across five datasets (Caltech101 [69], DTD [67], EuroSAT [65], FGVC-Aircraft [71], OxfordPets [73]) under the 16-shot setting and varying perturbation budgets $\epsilon = \{1/255, 2/255, 4/255\}$. Specifically, it assesses the sensitivity to different numbers of attack iterations $\iota = \{2, 4, 8\}$ and step sizes $\varsigma = \{\epsilon/\iota, 2\epsilon/\iota, 4\epsilon/\iota\}$. During the robustness evaluation phase, a 100-step PGD attack with the same perturbation budget and step size as in the training phase is employed to fully examine the defense capability of the model under strong attacks. We aim to determine the optimal combination of hyperparameters to more efficiently perform the next adversarial robustness tests under stronger attacks.

As evidenced in Table 5, employing larger attack step counts and step sizes during training ($\iota = 8, \varsigma = 4\epsilon/\iota$) does not enhance adversarial robustness during evaluation. Adversarial examples generated by PGD-8 tend to deviate significantly from the true data distribution, potentially causing the model to overfit the distribution of adversarial samples encountered during training rather than learning generalizable robust features. The model achieves higher natural accuracy when trained with a larger number of attack steps and a smaller step size ($\iota = 8, \varsigma = \epsilon/\iota$), as the resulting adversarial examples remain in close proximity to the original data manifold. The model demonstrates the capability to learn robust features while preserving discriminative power for natural samples. Across all perturbation budget settings, the combination of two attack iterations with a step size of $4\epsilon/\iota$ consistently achieves optimal robust accuracy and high clean accuracy. Therefore, we adopt this hyperparameter configuration for subsequent experiments involving varying perturbation budgets and different adversarial attack methods.

Table 5: Impact of perturbation budgets, attack iteration steps, and attack step sizes on natural and robust accuracy across 5 datasets under the 16-shot benchmark. Bold values highlight the best average results per perturbation budget.

| Pert. budg. $\epsilon$ | Iter. steps $\iota$ | Step size $\varsigma$ | Caltech101 Nat. | Caltech101 Rob. | DTD Nat. | DTD Rob. | EuroSAT Nat. | EuroSAT Rob. | FGVCAircraft Nat. | FGVCAircraft Rob. | OxfordPets Nat. | OxfordPets Rob. | Average Nat. | Average Rob. |
|---|---|---|---|---|---|---|---|---|---|---|---|---|---|---|
| 1/255 | 2 | $\epsilon/\iota$ | 94.16 | 89.57 | 65.37 | 54.20 | 86.42 | 71.67 | 38.94 | 25.47 | 87.05 | 70.84 | 74.39 | 62.35 |
| | | $2\epsilon/\iota$ | 94.04 | 89.78 | 65.84 | 56.03 | 84.77 | 70.60 | 39.39 | 27.51 | 86.29 | 71.85 | 74.07 | 63.15 |
| | | $4\epsilon/\iota$ | 94.20 | 90.55 | 65.19 | 57.03 | 86.35 | 73.52 | 39.39 | 28.74 | 87.33 | 74.35 | 74.49 | **64.84** |
| | 4 | $\epsilon/\iota$ | 94.36 | 89.86 | 65.43 | 55.38 | 86.16 | 71.20 | 39.18 | 26.52 | 87.44 | 70.43 | 74.51 | 62.68 |
| | | $2\epsilon/\iota$ | 94.08 | 89.78 | 65.13 | 55.38 | 85.62 | 73.37 | 39.24 | 27.24 | 86.37 | 71.74 | 74.09 | 63.50 |
| | | $4\epsilon/\iota$ | 94.04 | 90.14 | 65.66 | 56.21 | 86.47 | 74.75 | 39.72 | 27.30 | 86.86 | 72.47 | 74.55 | 64.18 |
| | 8 | $\epsilon/\iota$ | 94.20 | 89.74 | 66.31 | 54.85 | 86.52 | 69.32 | 39.30 | 25.59 | 87.11 | 69.80 | **74.69** | 61.86 |
| | | $2\epsilon/\iota$ | 94.04 | 89.98 | 65.72 | 55.56 | 86.49 | 73.10 | 38.94 | 28.05 | 86.07 | 71.41 | 74.25 | 63.62 |
| | | $4\epsilon/\iota$ | 94.24 | 89.90 | 65.19 | 55.50 | 86.65 | 74.09 | 38.88 | 27.18 | 86.59 | 71.95 | 74.31 | 63.72 |
| 2/255 | 2 | $\epsilon/\iota$ | 93.96 | 87.10 | 64.24 | 49.23 | 83.86 | 67.12 | 38.58 | 21.90 | 84.93 | 58.63 | 73.11 | 56.80 |
| | | $2\epsilon/\iota$ | 93.67 | 86.82 | 63.12 | 50.00 | 81.70 | 64.26 | 37.11 | 21.90 | 83.21 | 59.23 | 71.76 | 56.44 |
| | | $4\epsilon/\iota$ | 93.91 | 88.32 | 64.30 | 52.36 | 84.07 | 66.90 | 36.09 | 23.52 | 84.36 | 63.07 | 72.55 | **58.83** |
| | 4 | $\epsilon/\iota$ | 94.00 | 86.09 | 65.07 | 49.11 | 83.98 | 68.99 | 37.95 | 22.11 | 84.71 | 58.74 | **73.14** | 57.01 |
| | | $2\epsilon/\iota$ | 93.71 | 86.77 | 63.48 | 49.88 | 83.90 | 70.63 | 36.72 | 22.95 | 83.05 | 58.63 | 72.17 | 57.77 |
| | | $4\epsilon/\iota$ | 93.71 | 86.73 | 63.42 | 49.70 | 83.77 | 68.53 | 36.72 | 24.36 | 83.89 | 59.42 | 72.30 | 57.75 |
| | 8 | $\epsilon/\iota$ | 93.91 | 86.21 | 64.30 | 48.58 | 84.10 | 67.96 | 37.68 | 22.02 | 84.87 | 57.10 | 72.97 | 56.37 |
| | | $2\epsilon/\iota$ | 93.67 | 86.73 | 62.83 | 48.88 | 80.64 | 67.58 | 37.11 | 23.76 | 82.75 | 57.78 | 71.40 | 56.95 |
| | | $4\epsilon/\iota$ | 93.71 | 86.94 | 63.00 | 49.59 | 83.74 | 67.54 | 37.44 | 23.64 | 83.05 | 58.54 | 72.19 | 57.25 |
| 4/255 | 2 | $\epsilon/\iota$ | 92.41 | 82.15 | 60.87 | 41.08 | 79.53 | 67.17 | 33.39 | 17.28 | 79.26 | 40.88 | 69.09 | 49.71 |
| | | $2\epsilon/\iota$ | 92.01 | 80.41 | 58.92 | 38.36 | 79.54 | 57.63 | 32.67 | 18.24 | 74.79 | 39.60 | 67.59 | 46.85 |
| | | $4\epsilon/\iota$ | 92.58 | 83.20 | 59.63 | 45.27 | 79.49 | 58.51 | 33.33 | 19.80 | 79.45 | 49.03 | 68.90 | **51.16** |
| | 4 | $\epsilon/\iota$ | 92.74 | 81.30 | 61.82 | 40.07 | 80.12 | 64.15 | 33.57 | 17.76 | 79.50 | 41.40 | 69.55 | 48.94 |
| | | $2\epsilon/\iota$ | 92.01 | 80.20 | 59.69 | 40.31 | 78.70 | 63.54 | 31.53 | 17.85 | 75.61 | 39.55 | 67.51 | 48.29 |
| | | $4\epsilon/\iota$ | 92.33 | 80.45 | 59.04 | 38.42 | 79.57 | 59.01 | 33.03 | 18.54 | 76.04 | 39.11 | 68.00 | 47.11 |
| | 8 | $\epsilon/\iota$ | 93.10 | 81.99 | 61.76 | 40.07 | 82.06 | 60.85 | 35.07 | 18.15 | 78.30 | 40.47 | **70.06** | 48.31 |
| | | $2\epsilon/\iota$ | 91.60 | 79.63 | 59.46 | 40.19 | 78.99 | 61.49 | 32.37 | 18.60 | 75.28 | 38.35 | 67.54 | 47.65 |
| | | $4\epsilon/\iota$ | 91.85 | 80.81 | 58.92 | 40.60 | 78.51 | 63.32 | 32.82 | 19.50 | 76.02 | 40.94 | 67.62 | 49.03 |

## C.2 Impact of Perturbation Budget on Model Performance

We document the performance of CoAPT under four benchmark settings with three perturbation budgets $\epsilon = \{1/255, 2/255, 4/255\}$. The case of $\epsilon = 1/255$ corresponds to the results presented in the main text of the paper. As shown in Table 6 under the base-to-novel benchmark, the robust HM metrics decrease by 5.72% and 9.73% as the perturbation budgets increase, remaining within acceptable thresholds overall. The natural HM metrics decrease by only 2.27% and 4.59%, respectively, demonstrating the effectiveness of CoAPT in preserving natural generalization.

Table 6: Performance of CoAPT under varying perturbation budgets on the base-to-novel benchmark across 11 datasets.

| $\epsilon$ | | Metric | Caltech101 | DTD | EuroSAT | FGVCAircraft | Food101 | ImageNet | Flowers101 | OxfordPets | StanfordCars | SUN397 | UCF101 | Average |
|---|---|---|---|---|---|---|---|---|---|---|---|---|---|---|
| 1/255 | Nat. | Base | 97.25 | 76.08 | 91.61 | 35.37 | 78.20 | 66.15 | 94.94 | 90.55 | 73.34 | 76.69 | 82.95 | 78.47 |
| | | Novel | 92.72 | 54.03 | 56.11 | 25.41 | 79.47 | 55.41 | 63.07 | 94.50 | 59.20 | 70.46 | 68.45 | 65.35 |
| | | HM | 94.93 | 63.18 | 69.60 | 29.58 | 78.83 | 60.30 | 75.79 | 92.49 | 65.51 | 73.44 | 75.01 | 71.31 |
| | Rob. | Base | 94.38 | 67.98 | 84.67 | 25.37 | 62.03 | 52.65 | 88.57 | 78.72 | 54.20 | 64.50 | 71.65 | 67.70 |
| | | Novel | 88.03 | 43.88 | 47.40 | 16.68 | 62.86 | 45.07 | 51.89 | 83.71 | 40.95 | 58.50 | 56.50 | 54.13 |
| | | HM | 91.09 | 53.33 | 60.78 | 20.12 | 62.44 | 48.57 | 65.44 | 81.13 | 46.65 | 61.35 | 63.18 | 60.16 |
| 2/255 | Nat. | Base | 96.90 | 75.46 | 89.00 | 34.45 | 71.77 | 63.68 | 94.87 | 88.89 | 69.34 | 74.99 | 81.08 | 76.40 |
| | | Novel | 90.39 | 52.29 | 62.72 | 25.19 | 73.29 | 53.45 | 57.23 | 91.16 | 55.06 | 67.65 | 64.31 | 62.98 |
| | | HM | 93.53 | 61.78 | 73.58 | 29.11 | 72.52 | 58.12 | 71.40 | 90.01 | 61.38 | 71.13 | 71.72 | 69.04 |
| | Rob. | Base | 93.35 | 63.77 | 79.00 | 20.11 | 50.12 | 48.76 | 85.94 | 69.59 | 44.70 | 60.50 | 67.79 | 62.15 |
| | | Novel | 84.83 | 40.34 | 53.36 | 14.08 | 50.20 | 40.93 | 42.48 | 73.21 | 32.83 | 52.94 | 47.54 | 48.43 |
| | | HM | 88.88 | 49.42 | 63.70 | 16.53 | 50.16 | 44.50 | 56.86 | 71.35 | 37.86 | 56.47 | 55.89 | 54.44 |
| 4/255 | Nat. | Base | 95.22 | 71.99 | 89.24 | 29.65 | 64.67 | 58.48 | 91.17 | 84.26 | 62.12 | 71.27 | 77.40 | 72.32 |
| | | Novel | 86.24 | 48.31 | 65.10 | 22.14 | 64.41 | 48.42 | 49.08 | 85.51 | 47.78 | 63.70 | 58.73 | 58.13 |
| | | HM | 90.51 | 57.82 | 75.28 | 25.35 | 64.54 | 52.98 | 63.81 | 84.88 | 54.02 | 67.27 | 66.79 | 64.45 |
| | Rob. | Base | 88.32 | 54.75 | 74.52 | 15.97 | 34.56 | 39.71 | 78.63 | 54.12 | 32.03 | 51.19 | 57.08 | 52.81 |
| | | Novel | 75.11 | 32.97 | 50.56 | 10.62 | 31.62 | 33.04 | 30.92 | 57.10 | 23.32 | 43.99 | 37.21 | 38.77 |
| | | HM | 81.18 | 41.16 | 60.25 | 12.75 | 33.02 | 36.07 | 44.39 | 55.57 | 26.99 | 47.32 | 45.05 | 44.71 |

Table 7 reports the natural and robust accuracy of CoAPT under the 16-shot setting across different perturbation budgets. Compared to the base-to-novel setup, the few-shot scenario provides more training samples, enabling the model to exhibit greater stability when confronted with increased perturbations. Specifically, as the perturbation budgets increase, the robust accuracy declines by

5.42% and 8.73%, while the natural accuracy drops by only 2.26% and 4.42%, indicating a more moderate performance degradation trend.

Table 7: Performance of CoAPT under varying perturbation budgets on the few-shot benchmark across 11 datasets.

| $\epsilon$ | Metric | Caltech101 | DTD | EuroSAT | FGVCAircraft | Food101 | ImageNet | Flowers101 | OxfordPets | StanfordCars | SUN397 | UCF101 | Average |
|---|---|---|---|---|---|---|---|---|---|---|---|---|---|
| 1/255 | Nat. | 94.51 | 65.50 | 85.42 | 38.60 | 73.00 | 62.96 | 93.86 | 86.82 | 73.94 | 70.84 | 79.09 | 74.96 |
| | Rob. | 90.03 | 56.09 | 71.43 | 27.39 | 56.72 | 51.32 | 85.36 | 72.28 | 55.91 | 58.33 | 67.91 | 62.98 |
| 2/255 | Nat. | 93.91 | 64.30 | 84.07 | 36.09 | 68.08 | 61.38 | 91.64 | 84.36 | 70.25 | 69.42 | 76.24 | 72.70 |
| | Rob. | 88.48 | 52.36 | 66.90 | 23.28 | 46.99 | 47.79 | 81.20 | 63.07 | 47.05 | 54.52 | 61.56 | 57.56 |
| 4/255 | Nat. | 92.58 | 59.63 | 79.49 | 33.33 | 60.69 | 56.98 | 87.74 | 79.45 | 63.50 | 65.70 | 72.03 | 68.28 |
| | Rob. | 83.20 | 45.27 | 58.51 | 20.10 | 32.19 | 40.14 | 72.55 | 49.03 | 35.43 | 47.11 | 53.56 | 48.83 |

As shown in the evaluation results under the zero-shot settings in Table 8, our model consistently demonstrates strong natural generalization, adversarial robustness, and stability across different perturbation budgets. Specifically, under the zero-shot scenario, the average robust accuracy decreases by 3.93% and 7.35% with increasing perturbation budgets, while the average natural accuracy declines by only 1.63% and 4.21%. The results indicate that the model maintains strong perturbation resistance even under extreme generalization conditions. The evaluation results under the out-of-distribution settings in Table 9 exhibit a similar trend.

Table 8: Performance of CoAPT under varying perturbation budgets on the zero-shot benchmark across 11 datasets.

| $\epsilon$ | Metric | ImageNet | Caltech101 | DTD | EuroSAT | FGVCAircraft | Food101 | Flowers101 | OxfordPets | StanfordCars | SUN397 | UCF101 | Average |
|---|---|---|---|---|---|---|---|---|---|---|---|---|---|
| 1/255 | Nat. | 63.42 | 89.10 | 36.66 | 28.37 | 13.84 | 69.85 | 56.30 | 82.98 | 47.99 | 58.10 | 57.34 | 54.91 |
| | Rob. | 51.18 | 83.29 | 30.59 | 20.13 | 9.43 | 55.99 | 45.34 | 70.03 | 30.24 | 46.64 | 47.36 | 44.57 |
| 2/255 | Nat. | 61.33 | 88.84 | 36.52 | 26.94 | 11.64 | 66.13 | 55.42 | 82.07 | 46.14 | 56.52 | 54.51 | 53.28 |
| | Rob. | 46.98 | 80.24 | 29.31 | 19.06 | 7.17 | 50.28 | 41.29 | 64.54 | 23.96 | 42.11 | 42.11 | 40.64 |
| 4/255 | Nat. | 56.77 | 87.42 | 33.75 | 22.72 | 12.39 | 57.69 | 48.40 | 77.13 | 41.01 | 52.60 | 49.85 | 49.07 |
| | Rob. | 38.62 | 75.66 | 25.24 | 14.17 | 6.51 | 36.61 | 33.82 | 52.03 | 15.00 | 34.87 | 33.65 | 33.29 |

Table 9: Performance of CoAPT under varying perturbation budgets on the out-of-distribution benchmark across 11 datasets.

| $\epsilon$ | ImageNet-A | | ImageNet-R | | ImageNet-Sketch | | ImageNet-V2 | | Average | |
|---|---|---|---|---|---|---|---|---|---|---|
| | Nat. | Rob. | Nat. | Rob. | Nat. | Rob. | Nat. | Rob. | Nat. | Rob. |
| 1/255 | 16.99 | 9.72 | 60.35 | 50.71 | 35.76 | 29.13 | 54.35 | 42.23 | 41.86 | 32.95 |
| 2/255 | 14.27 | 7.49 | 57.81 | 45.92 | 34.26 | 25.81 | 52.54 | 38.50 | 39.72 | 29.43 |
| 4/255 | 10.35 | 4.23 | 54.28 | 38.80 | 32.17 | 21.10 | 47.63 | 30.67 | 36.11 | 23.70 |

## C.3 Robustness Evaluation under Varying Attacks

We evaluate our method using attack types based on different perturbation mechanisms. The CW attack is an optimization-based method designed to generate adversarial perturbations that are minimal in magnitude yet highly effective in misleading the model. It has demonstrated strong attack performance across various tasks. The TPGD attack is a targeted variant of the PGD attack that misdirects samples toward specific target classes. AutoAttack is an ensemble-based, parameter-free robustness evaluation framework that integrates multiple strong attack algorithms to provide reliable adversarial assessment results. Specifically, we evaluate CW, TPGD, and AutoAttack attacks under the zero-shot benchmark, while only CW and TPGD are evaluated under the base-to-novel benchmark. We adopt PGD attack with the hyperparameter configuration $\epsilon = 4/255$, $\iota = 2$, $\varsigma = 4\epsilon/\iota$ for adversarial training. During the robustness evaluation phase, both CW and TPGD attacks are applied with the same perturbation budget and step size, while the number of attack steps is uniformly set to 100. For AutoAttack, we use the same perturbation budget ($\epsilon = 4/255$), and its attack process does not rely on hyperparameters such as step size or the number of steps. Overall, the robustness advantage of our method is not a result of overfitting to any specific attack.

Table 10: Performance of CoAPT against various attack methods under the base-to-novel benchmark.

| Type | Metric | | Caltech101 | DTD | EuroSAT | FGVCAircraft | Food101 | ImageNet | Flowers101 | OxfordPets | StanfordCars | SUN397 | UCF101 | Average |
|---|---|---|---|---|---|---|---|---|---|---|---|---|---|---|
| CW | Nat. | Base | 95.22 | 71.99 | 89.24 | 29.65 | 64.67 | 58.48 | 91.17 | 84.26 | 62.12 | 71.27 | 77.40 | 72.32 |
| | | Novel | 86.24 | 48.31 | 65.10 | 22.14 | 64.41 | 48.42 | 49.08 | 85.51 | 47.78 | 63.70 | 58.73 | 58.13 |
| | | HM | 90.51 | 57.82 | 75.28 | 25.35 | 64.54 | 52.98 | 63.81 | 84.88 | 54.02 | 67.27 | 66.79 | 64.45 |
| | Rob. | Base | 86.38 | 59.38 | 81.64 | 21.67 | 58.12 | 51.32 | 86.32 | 67.68 | 44.50 | 60.27 | 67.68 | 62.27 |
| | | Novel | 75.00 | 38.41 | 51.64 | 17.34 | 56.55 | 40.39 | 42.48 | 69.35 | 35.31 | 52.56 | 46.73 | 47.80 |
| | | HM | 80.29 | 46.64 | 63.27 | 19.26 | 57.33 | 45.20 | 56.94 | 68.50 | 39.37 | 56.15 | 55.29 | 54.08 |
| TPGD | Nat. | Base | 95.16 | 71.99 | 89.21 | 29.83 | 64.68 | 58.46 | 91.17 | 84.26 | 62.12 | 71.21 | 77.40 | 72.32 |
| | | Novel | 86.24 | 48.31 | 65.05 | 22.08 | 64.39 | 48.44 | 49.01 | 85.51 | 47.81 | 63.66 | 58.73 | 58.11 |
| | | HM | 90.48 | 57.82 | 75.24 | 25.37 | 64.53 | 52.98 | 63.75 | 84.88 | 54.03 | 67.22 | 66.79 | 64.44 |
| | Rob. | Base | 93.74 | 68.29 | 90.62 | 29.59 | 62.93 | 55.85 | 90.50 | 79.59 | 58.12 | 68.74 | 75.85 | 70.35 |
| | | Novel | 84.39 | 42.39 | 62.41 | 21.60 | 62.67 | 46.47 | 48.01 | 80.48 | 43.85 | 61.42 | 58.36 | 55.64 |
| | | HM | 88.82 | 52.31 | 73.91 | 24.97 | 62.80 | 50.73 | 62.74 | 80.03 | 49.99 | 64.88 | 65.96 | 62.14 |

As can be seen from the experimental results under the base-to-novel benchmark in Table 10, our approach exhibits strong robust generalization capabilities when confronted with different types of adversarial attacks. Overall, the CW attack is more destructive. Although it induces significant accuracy degradation on novel classes, the performance remains within acceptable range. In contrast, under the TPGD attack, the model maintains relatively high natural and robust accuracy, further validating the stable performance of CoAPT across different types of adversarial attacks.

Figure 4 presents the robust accuracy of the model under CW, AutoAttack, and TPGD attacks across 11 datasets in the zero-shot benchmark. In terms of overall trends, the model demonstrates the strongest robustness under TPGD attacks, achieving the highest robust accuracy across nearly all datasets. In contrast, CW attacks are more destructive, particularly showing stronger attack effectiveness on complex datasets such as ImageNet and StanfordCars. AutoAttack, as an ensemble-based evaluation framework, displays intermediate attack strength between CW and TPGD. Moreover, significant robustness variations exist across different datasets. The model maintains relatively high robust accuracy on Caltech101, Flowers102, and OxfordPets, while showing noticeably lower performance on FGVCAircraft and EuroSAT.

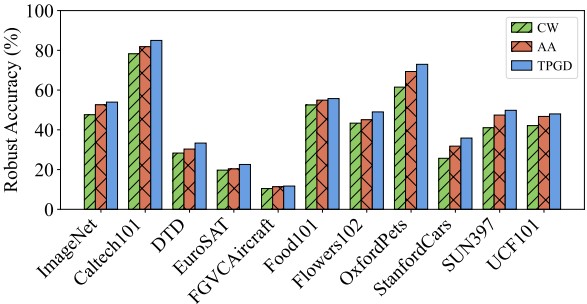

Figure 4: Comparison of robust accuracy under different attack methods on zero-shot benchmarks.

To evaluate the impact of $\ell_2$-norm adversarial attacks on robust VLMs, we designed and conducted an experiment based on $\ell_2$-norm perturbations. The training weights were derived from the $\ell_\infty$-based PGD attack, and the evaluation settings remained consistent. Table 11 presents the experimental results of our approach across five datasets under varying perturbation budgets. It can be observed that as the perturbation budget increases, the model's classification accuracy experiences a moderate decline. Nevertheless, our approach significantly improves the model's robustness against $\ell_2$-norm attacks, even under the $\ell_\infty$-norm threat model.

Table 11: Robust accuracy under $\ell_2$-norm PGD attacks on the base-to-novel benchmark.

| $\epsilon$ | Caltech101 | | DTD | | EuroSAT | | FGVCAircraft | | OxfordPets | |
|---|---|---|---|---|---|---|---|---|---|---|
| | Base | Novel | Base | Novel | Base | Novel | Base | Novel | Base | Novel |
| 1/255 | 94.25 | 87.99 | 68.29 | 44.44 | 85.45 | 55.23 | 25.87 | 16.86 | 80.75 | 83.95 |
| 2/255 | 92.32 | 84.06 | 62.38 | 37.80 | 79.64 | 54.49 | 20.05 | 12.96 | 73.52 | 75.17 |
| 4/255 | 90.70 | 81.00 | 61.00 | 36.96 | 81.00 | 54.38 | 18.91 | 13.92 | 70.28 | 70.86 |

## C.4 Sensitivity Analysis of Prompt Length and Depth in Multimodal Prompting

**Prompt depth and prompt length.** We conduct ablation studies on prompt depth and prompt length under the base-to-novel setting across 10 datasets, excluding ImageNet and its variants. Figure 5 summarizes the average results over these datasets. As shown in the left panel of Figure 5, model performance steadily improves with increasing adversarial prompt depth. However, performance gains plateau when the depth exceeds nine layers, showing diminishing returns. To avoid introducing excessive trainable parameters, we ultimately set the prompt depth to 9.

The right panel of Figure 5 illustrates the impact of prompt length on model performance. As the number of prompt tokens increases, the natural and robust performance on base classes remains relatively stable, whereas the natural and robust performance of the novel classes exhibits a declining trend. This indicates that excessive trainable prompt tokens are prone to overfit task-specific features, thereby undermining the task-agnostic generalization capability of VLMs. Similar performance trends have also been reported in the literature [57]. The model achieves optimal performance when the prompt length is set to 4.

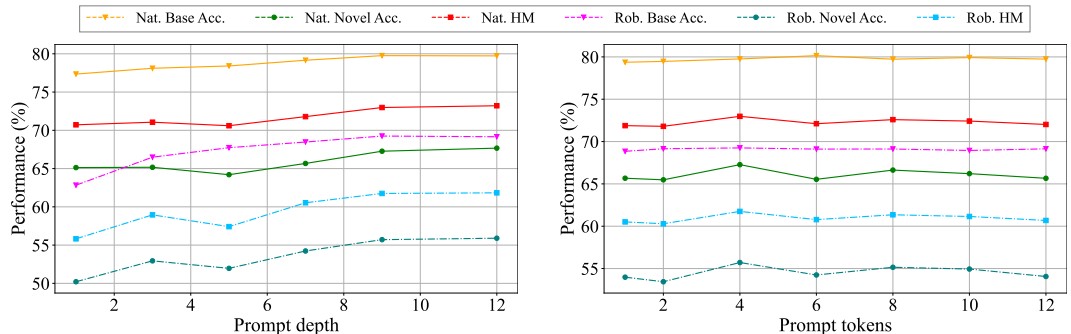

Figure 5: Analyze the impact of prompt depth (*left*) and prompt length (*right*) on the performance.

## C.5 Performance Across Different CLIP Architectures

We additionally evaluate CoAPT on the CLIP ViT-B/16 architecture under the base-to-novel benchmark to verify its scalability to higher-resolution architectures in terms of both natural accuracy and adversarial robustness. Compared to ViT-B/32, the ViT-B/16 architecture adopts finer image patching granularity, resulting in a greater number of input tokens and consequently exhibiting superior spatial resolution representation capacity. This structural advantage typically leads to enhanced performance in fine-grained visual tasks.

Table 12: Results of base-to-novel benchmarks on the ViT-B/16 architecture of CLIP under 11 datasets.

| | Metric | Caltech101 | DTD | EuroSAT | FGVCAircraft | Food101 | ImageNet | Flowers101 | OxfordPets | StanfordCars | SUN397 | UCF101 | Average |
|---|---|---|---|---|---|---|---|---|---|---|---|---|---|
| CLIP ViT-B/16 Nat. | Base | 97.93 | 78.13 | 93.64 | 41.54 | 83.26 | 71.76 | 97.63 | 94.10 | 77.49 | 79.21 | 83.82 | 81.68 |
| | Novel | 94.00 | 56.40 | 54.36 | 32.87 | 83.50 | 60.02 | 67.09 | 94.69 | 63.33 | 72.96 | 72.36 | 68.33 |
| | HM | 95.92 | 65.51 | 68.79 | 36.70 | 83.38 | 65.36 | 79.53 | 94.39 | 69.70 | 75.95 | 77.67 | 74.41 |
| CLIP ViT-B/16 Rob. | Base | 96.45 | 71.76 | 90.05 | 34.21 | 72.14 | 61.72 | 92.31 | 87.08 | 62.24 | 70.66 | 75.28 | 73.99 |
| | Novel | 90.72 | 51.45 | 47.64 | 25.07 | 72.43 | 52.47 | 58.09 | 88.59 | 48.30 | 64.33 | 63.17 | 60.21 |
| | HM | 93.50 | 59.93 | 62.31 | 28.94 | 72.28 | 56.72 | 71.30 | 87.83 | 54.40 | 67.35 | 68.70 | 66.39 |

Compared to the CLIP ViT-B/32 results reported in Table 1 of the main text, Table 12 demonstrates that the CLIP ViT-B/16 architecture achieves improvements of 3.1% and 6.23% in the HM of natural accuracy and robust accuracy, respectively. The high-resolution visual representations of the ViT-B/16 architecture provide CoAPT with a finer-grained and more stable latent space, enabling more effective reconstruction of natural generalization features disrupted by adversarial perturbations. Compared to the ViT-B/32 architecture, this enhanced representational capacity mitigates alignment errors and distributional shifts between language and vision embeddings, thereby significantly improving the natural generalization and adversarial robustness of robust CLIP. In contrast, the FAP method fails to achieve robustness gains under the ViT-B/16 architecture, further demonstrating the superiority of CoAPT in terms of scalability and stability.

## C.6 Impact of Reconstruction Loss Functions on Model Performance

CoAPT employs a Gaussian radial basis function (RBF) to measure the similarity between the language and vision branch embeddings of natural and robust CLIP representations in the latent space, effectively capturing the impact of input perturbations on the feature distributions. In Table 13, we systematically compare the performance of CoAPT on the base-to-novel benchmark under different configurations of Gaussian RBF and standard MSE loss functions. The Gaussian RBF demonstrates absolute superiority over MSE by 5.15% and 4.81% in natural HM and robust HM metrics, respectively. This is attributed to the fact that Gaussian RBF can effectively amplify the feature shifts caused by small-scale perturbations to acutely capture the subtle distributional changes, which not only promotes robustness training but also inhibits overfitting to a certain extent.

## C.7 Independent and Joint Vision-Language Adversarial Prompting

CoAPT employs deep contextualized joint vision-language adversarial prompting (JVLAP), which refines visual prompts based on linguistic prompts to facilitate cross-modal co-optimization via a

Table 13: Results of base-to-novel benchmarks using Gaussian RBF and MSE loss functions under 11 datasets.

| | Metric | Caltech101 | DTD | EuroSAT | FGVCAircraft | Food101 | ImageNet | Flowers101 | OxfordPets | StanfordCars | SUN397 | UCF101 | Average |
|---|---|---|---|---|---|---|---|---|---|---|---|---|---|
| Gauss RBF — Nat. | Base | 97.25 | 76.08 | 91.61 | 35.37 | 78.20 | 66.15 | 94.94 | 90.55 | 73.34 | 76.69 | 82.95 | 78.47 |
| | Novel | 92.72 | 54.03 | 56.11 | 25.41 | 79.47 | 55.41 | 63.07 | 94.50 | 59.20 | 70.46 | 68.45 | 65.35 |
| | HM | 94.93 | 63.18 | 69.60 | 29.58 | 78.83 | 60.30 | 75.79 | 92.49 | 65.51 | 73.44 | 75.01 | 71.31 |
| Gauss RBF — Rob. | Base | 94.38 | 67.98 | 84.67 | 25.37 | 62.03 | 52.65 | 88.57 | 78.72 | 54.20 | 64.50 | 71.65 | 67.70 |
| | Novel | 88.03 | 43.88 | 47.40 | 16.68 | 62.86 | 45.07 | 51.89 | 83.71 | 40.95 | 58.50 | 56.50 | 54.13 |
| | HM | 91.09 | 53.33 | 60.78 | 20.12 | 62.44 | 48.57 | 65.44 | 81.13 | 46.65 | 61.35 | 63.18 | 60.16 |
| MSE — Nat. | Base | 96.26 | 73.50 | 94.29 | 33.97 | 72.59 | 62.11 | 94.97 | 89.10 | 71.34 | 73.24 | 79.63 | 76.45 |
| | Novel | 89.30 | 46.62 | 41.03 | 21.30 | 74.55 | 46.00 | 53.97 | 91.50 | 49.94 | 64.65 | 62.52 | 58.31 |
| | HM | 92.65 | 57.05 | 57.17 | 26.18 | 73.56 | 52.85 | 68.83 | 90.28 | 58.75 | 68.67 | 70.04 | 66.16 |
| MSE — Rob. | Base | 93.35 | 64.35 | 86.24 | 24.19 | 56.18 | 48.90 | 89.36 | 76.50 | 53.30 | 61.17 | 70.94 | 65.86 |
| | Novel | 85.15 | 38.04 | 34.46 | 14.04 | 57.21 | 36.04 | 41.91 | 80.76 | 34.07 | 52.12 | 51.22 | 47.73 |
| | HM | 89.06 | 47.82 | 49.24 | 17.77 | 56.69 | 41.50 | 57.06 | 78.57 | 41.57 | 56.28 | 59.49 | 55.35 |

vision-language coupling network. In Table 14, we additionally report the performance of CoAPT using independent vision-language adversarial prompting (IVLAP) under the base-to-novel benchmark. Compared to the JVLAP results in Table 1, IVLAP exhibits reductions of 0.29% and 0.37% in the HM of natural and robust accuracy, respectively. Although IVLAP shows slightly better performance on the Flowers101 and StanfordCars datasets, its performance on most other datasets is comparable to or slightly inferior to that of JVLAP.

Table 14: Performance of CoAPT using the IVLAP scheme on 11 datasets under the base-to-novel benchmark.

| | Metric | Caltech101 | DTD | EuroSAT | FGVCAircraft | Food101 | ImageNet | Flowers101 | OxfordPets | StanfordCars | SUN397 | UCF101 | Average |
|---|---|---|---|---|---|---|---|---|---|---|---|---|---|
| IVLAP — Nat. | Base | 96.71 | 76.74 | 92.88 | 33.91 | 78.32 | 66.30 | 95.73 | 90.86 | 72.24 | 76.98 | 81.13 | 78.34 |
| | Novel | 92.25 | 53.02 | 52.41 | 26.75 | 80.29 | 55.20 | 64.18 | 94.13 | 58.83 | 70.31 | 67.01 | 64.94 |
| | HM | 94.43 | 62.71 | 67.01 | 29.91 | 79.29 | 60.24 | 76.84 | 92.46 | 64.85 | 73.50 | 73.40 | 71.02 |
| IVLAP — Rob. | Base | 94.58 | 67.71 | 84.88 | 24.67 | 61.22 | 52.57 | 89.55 | 79.11 | 53.87 | 64.63 | 69.39 | 67.47 |
| | Novel | 88.10 | 43.12 | 43.87 | 16.74 | 63.57 | 44.94 | 52.06 | 83.84 | 41.67 | 58.73 | 53.92 | 53.69 |
| | HM | 91.22 | 52.68 | 57.85 | 19.94 | 62.37 | 48.46 | 65.84 | 81.40 | 46.99 | 61.54 | 60.69 | 59.79 |

JVLAP shows more significant advantages in modeling cross-modal robustness. By jointly optimizing adversarial features of both vision and language branches within a unified framework, it more effectively captures the synergistic variations between the two modalities in the latent space, thereby enhancing the consistency and stability of modality alignment. This joint optimization not only mitigates performance bias caused by asymmetrical perturbation sensitivity between modalities but also preserves semantic consistency during adversarial training. Consequently, it significantly enhances the generalization capability of the model on novel categories, zero-shot recognition, and out-of-distribution scenarios.

# D  Impact Statement

This work aims to support progress in robust machine learning by improving the resilience of vision-language models against adversarial threats. Although we do not anticipate any immediate negative consequences, it is important to remain aware of potential misuse in security-critical domains. One key outcome of our approach is the ability to preserve robustness with low-cost model adjustments, which offers practical value for time-sensitive applications on mobile and resource-limited devices. The techniques introduced here may contribute to safer and more dependable deployment of AI systems in real-world environments, particularly in areas like intelligent sensing and mobile security.

# E  Reproducibility

To support reproducibility, we have included the anonymized source code in the supplementary materials for the review process. If the paper is accepted, we will release the complete codebase to the public.

# F  Limitations

This work primarily investigates adversarial robustness against image-level perturbations, while multi-modal adversarial attacks that simultaneously affect both vision and language inputs remain

underexplored. The current framework assumes that adversarial noise originates solely from the visual modality, which limits its applicability in scenarios involving adversarial manipulations in textual inputs. Although the proposed latent space reconstruction method shows strong generalization in experiments, its specific impact on generalization behavior and the theoretical analysis for its superiority over other techniques remain unexplained. The influence of latent space structure and distribution on model robustness and generalization requires further theoretical exploration. We leave these limitations as essential directions for future investigation.

