# OpenReview forum: "Learning Robust Vision-Language Models from Natural Latent Spaces"
_NeurIPS.cc/2025/Conference — NeurIPS 2025 spotlight_

### Official Review · Reviewer_2CcH · 2025-06-13

**Clarity:** 3
**Significance:** 4
**Originality:** 3
**Rating:** 5
**Confidence:** 4

**Summary:**

This work presents a novel adversarial prompt tuning framework through a collaborative process. It proposes the adoption of a real-time Total Variation regularization algorithm to suppress high-frequency noise that undermines adversarial robustness, followed by the reconstruction of natural features using the original CLIP model.

**Questions:**

1. From the ablation results in Table 4, it appears that Adp. FGP plays a critical role in the entire framework. What contributes to its importance in your proposed method, and have you considered alternative denoising methods?

**Ethical Concerns:**

["NO or VERY MINOR ethics concerns only"]

**Final Justification:**

The authors gave positive response to my questions.

I think the initial assessment remains a fair evaluation. I will keep my score.

**Limitations:**

yes

**Quality:**

3

**Strengths And Weaknesses:**

Strength：

- The method provides a new workflow for addressing adversarial prompt tuning tasks, which is quite innovative. It adopts a two-step pipeline: high-frequency noise suppression followed by natural feature reconstruction. Task-specific improvements are proposed for each sub-process based on the characteristics of the task.

- The experimental results are compelling, significantly outperforming previous SOTA methods.

- The strategy of imposing constraints between natural and adversarial features in adversarial prompt learning is particularly insightful. Compared to baseline methods, the proposed loss design is more systematic and demonstrates better numerical stability, enabling more effective feature alignment and reconstruction.

Weakness：

- The quality of this work could be further improved by clearly stating the motivation behind each module design. For instance, the authors introduced the Low-Rank Residual Module and Rényi Regularization in Section 3.3, but did not explain how these specific components or loss terms benefit the task.

- The authors reported only performance metrics, without including any efficiency-related indicators (e.g., GPU usage, training time). For few-shot and zero-shot prompt learning tasks, the efficiency of downstream adaptation is also a key factor influencing the practical applicability of the algorithm.

---

> ### Author Rebuttal · Authors · 2025-07-31
>
> **We thank reviewer for the constructive comments. We provide our feedback as follows. This will ultimately be reflected in the revision of the paper.**
>
>
> **Answer for W1:**
> We introduce Rényi divergence as a regularization term in the latent space, aiming to enhance the stability of the model during robust training. As a parametric instance of f-divergence, Rényi divergence, when applied to surrogate losses, enables adaptive modulation of the model's sensitivity to regions with varying probability densities by tuning the parameter $\alpha$, thereby flexibly adjusting the emphasis on dominant modes and anomalous samples. In contrast, the standard KL divergence imposes disproportionate penalties on minor deviations between the model and target distributions in the tail regions, resulting in rapid gradient amplification that triggers training instability and overfitting. When $\alpha > 1$, the $\alpha$-loss induced by Rényi divergence exhibits milder gradient behavior in low-density regions, guiding the model to concentrate on high-density principal modes and thereby yielding a smoother and more controllable optimization trajectory [1,2]. The low-rank residual module constrains the redundant degrees of freedom in the feature space, effectively constructing a degenerate yet discriminative subspace representation. It enables the model to retain backbone features while selectively learning task-specific fine-grained deviations within a compact subspace.
>
>
> **Answer for W2 & Q1:**
> We focus on denoising methods with real-time capability and evaluate several classical approaches in terms of both performance and computational efficiency. As shown in Table 1, we compare the total runtime for 16-shot training and testing on the DTD dataset using an NVIDIA Tesla V100 GPU. The proposed adaptive-FGP method employs 10 iterative steps. Under this setting, the time overhead increases by only 10.7\% compared to the baseline model without contrastive mechanisms. Given the observed performance gains, this overhead remains acceptable. For other classical baselines, wavelet denoising and the Split Bregman method are implemented via the official skimage library, while TVM adopts a precompiled C++ dynamic library for Total Variation denoising. Notably, our adaptive-FGP method incorporates a parallelized implementation that runs directly on CUDA, leading to significantly lower runtime compared to others. However, these runtime differences do not directly reflect computational complexity. Table 2 further reports the average, minimum, and maximum per-image inference time of adaptive-FGP, measured after one epoch over the entire training set.
>
> In terms of accuracy, our method achieves state-of-the-art performance on both clean and robust evaluations,  particularly excelling in robustness. By integrating a spatially adaptive regularization strategy and an accelerated gradient method with an adaptive restart mechanism, we improve the traditional TV approach to simultaneously enhance performance and real-time capability.
>
> **Table 1. Denoising Runtime and Accuracy (Natural vs. Adversarial)**
>
> | Method                    | Nat (%) | Rob (%) | Average (%) | Runtime (s) | Time Increase (%) |
> |--------------------------|---------|---------|--------------|--------------|--------------------|
> | No Denoising  | 64.3    | 27.8    | 46.05        | 1875         | ~                  |
> | Wavelet Denoising        | 64.2    | 28.8    | 46.50        | 2913         | 55.4%              |
> | Split Bregman            | 64.7    | 31.6    | 48.15        | 2345         | 25.1%              |
> | TVM (Guo et al.[3])      | 56.9    | 50.8    | 53.85        | 7983         | 325.8%             |
> | Adaptive-FGP             | 66.1    | 53.0    | 59.55        | 2075         | 10.7%              |
>
> **Table 2. Adaptive-FGP Latency per Image (ms)**
>
> | Max Time | Min Time | Average Time |
> |----------|----------|--------------|
> |   5.72   |   2.81   |     3.71     |
>
> **Answer for Q1:**
> CoAPT can be viewed as a generalized formulation of masked image modeling (MIM). We transform the block-level masking in conventional MIM into fine-grained, pixel-level denoising, and redefine the reconstruction objective as the latent representations of the pretrained CLIP model. The adaptive-FGP module mitigates adversarial perturbations by suppressing high-frequency features, at the cost of degrading image details. Masked image modeling enhances generalization to downstream tasks by learning high-level representations during the image reconstruction process. However, its sensitivity to high-frequency features exposes vulnerabilities to adversarial perturbations. CoAPT’s adaptive-FGP module alleviates this weakness by attenuating high-frequency adversarial components. Although adaptive-FGP induces degradation in natural image content, such distortions can be effectively recovered in the latent space through MIM, which tolerates strong denoising or even structural corruption. As a result, the limitations of each component are mutually compensated, enabling their respective strengths in adversarial robustness and generalization to be jointly maximized.
>
> Therefore, the coupling between adaptive-FGP and image reconstruction is essential to performance improvement. When adaptive-FGP is entirely removed and adversarial images are directly encoded for latent-space reconstruction, the inherent modeling mechanism of MIM leads it to overly focus on high-frequency regions that are heavily perturbed. Although some robustness gains remain, they are significantly compromised by the influence of high-frequency adversarial noise. Consequently, removing adaptive-FGP not only eliminates its inherent denoising capability but also disrupts its synergistic interaction with MIM. This decoupling exposes MIM’s tendency to emphasize high-frequency artifacts, ultimately resulting in degraded performance.
>
> [1] Kurri, Gowtham R., et al. "α-GAN: Convergence and estimation guarantees." 2022 IEEE International Symposium on Information Theory (ISIT). IEEE, 2022.
> [2] Sypherd, Tyler, et al. "A tunable loss function for robust classification: Calibration, landscape, and generalization." IEEE Transactions on Information Theory 68.9 (2022): 6021-6051.
> [3]Guo, C., et al. Countering adversarial images using input transformations. ICLR 2018.

---

> ### Comment · Reviewer_2CcH · 2025-08-05
> **Official Comment by Reviewer 2CcH**
>
> The authors gave positive response to my questions.
>
> I think the initial assessment remains a fair evaluation. I will keep my score to vote for acceptance.

---

> > ### Author Response · Authors · 2025-08-05
> >
> > We sincerely appreciate your insightful comments and positive feedback. These suggestions are instrumental in refining the arguments presented and improving the overall quality of our manuscript.

---

### Official Review · Reviewer_Uzgu · 2025-06-18

**Clarity:** 3
**Significance:** 3
**Originality:** 3
**Rating:** 5
**Confidence:** 2

**Summary:**

This paper proposes Collaborative Adversarial Prompt Tuning (CoAPT), a novel method for enhancing the adversarial robustness of Vision-Language Models (VLMs) in their shared latent space for natural images and text. Unlike existing Prompt Tuning methods, CoAPT utilizes an improved real-time Total Variation regularization to suppress high-frequency details in images, thereby reducing the effect of adversarial noise. Furthermore, it introduces a residual module with a loss for representation restoration and incorporates Renyi divergence to minimize the gap between the distributions of natural and adversarial examples. The proposed method demonstrates superior performance over existing approaches in few-shot learning, generalization to new classes, zero-shot, and out-of-distribution (OOD) scenarios, achieving an optimal balance between robustness and generalization.

**Questions:**

- Are there results for other norm-based attacks (e.g., an L2 attack)? I am curious to know if training against an Linf threat model also confers robustness against L2 attacks.
- Is the proposed method applicable to other types of models, such as those with a ResNet backbone? If so, I am curious to see if it yields similar results.
- Can this method be applied to a multi-modal LLM (e.g., LLaVA) without issues? While it appears effective for the vision-language tasks presented, I am curious whether there are any problems from the perspective of output feature drift when this robust model is used as a vision encoder within a larger system.

**Ethical Concerns:**

["NO or VERY MINOR ethics concerns only"]

**Final Justification:**

I am raising my score based on the authors' rebuttal. My initial review highlighted reservations regarding the method's practical overhead and generalizability, but the authors have fully addressed these concerns. The newly added experiments provided by the authors are exemplary. They convincingly illustrate that the method's effectiveness is not limited to specific threat models or architectures initially evaluated. Furthermore, by showing its successful application as a vision encoder within a generative multi-modal LLM, they resolve my concerns about its scalability and potential side effects. This additional evidence resolves my primary reservations. It clearly demonstrates that the proposed method is effective, versatile, and robust.

**Limitations:**

See weaknesses and questions

**Quality:**

2

**Strengths And Weaknesses:**

Strengths

This paper effectively solves the robust-accuracy trade-off, a persistent issue when considering the adversarial robustness of vision-language models. In particular, beyond simply using natural-adversarial examples, it utilizes an adaptive-FGP that considers both the characteristics of adversarial noise and the shortcomings of existing methodologies. Regarding the training loss, instead of merely updating the model with a reconstruction loss, it introduces a residual module inspired by LoRA, which helps mitigate the trade-off. Additionally, it further improves performance by introducing Renyi regularization. As a result, the method achieves high performance on both clean and robust metrics across various datasets and tasks compared to existing baselines.

Weaknesses

The additional computations, including the proposed real-time TV regularization and the restoration loss in the latent space, increase the model's complexity during training and inference, which could lead to higher computational costs in practical applications. While this may not seem like a major issue in the current few-shot setting, difficulties could arise when scaling the methodology to large-scale datasets. Additionally, it is difficult to discern the specific effect of each applied component. Although the authors indirectly address this through an ablation study, it is still hard to determine if any redundant terms exist in the framework.

---

> ### Author Rebuttal · Authors · 2025-07-31
>
> **We thank reviewer for the constructive comments. We provide our feedback as follows. This will ultimately be reflected in the revision of the paper.**
>
>
> **Answer for W1:**
> We focus on denoising methods with real-time capability and evaluate several classical approaches in terms of both performance and computational efficiency. As shown in Table 1, we compare the total runtime for 16-shot training and testing on the DTD dataset using an NVIDIA Tesla V100 GPU. The proposed adaptive-FGP method employs 10 iterative steps. Under this setting, the time overhead increases by only 10.7\% compared to the baseline model without contrastive mechanisms. Given the observed performance gains, this overhead remains acceptable. For other classical baselines, wavelet denoising and the Split Bregman method are implemented via the official skimage library, while TVM adopts a precompiled C++ dynamic library for Total Variation denoising. Notably, our adaptive-FGP method incorporates a parallelized implementation that runs directly on CUDA, leading to significantly lower runtime compared to others. However, these runtime differences do not directly reflect computational complexity. Table 2 further reports the average, minimum, and maximum per-image inference time of adaptive-FGP, measured after one epoch over the entire training set.
>
> In terms of accuracy, our method achieves state-of-the-art performance on both clean and robust evaluations,  particularly excelling in robustness. By integrating a spatially adaptive regularization strategy and an accelerated gradient method with an adaptive restart mechanism, we improve the traditional TV approach to simultaneously enhance performance and real-time capability.
>
> **Table 1. Denoising Runtime and Accuracy (Natural vs. Adversarial)**
>
> | Method                    | Nat (%) | Rob (%) | Average (%) | Runtime (s) | Time Increase (%) |
> |--------------------------|---------|---------|--------------|--------------|--------------------|
> | No Denoising  | 64.3    | 27.8    | 46.05        | 1875         | ~                  |
> | Wavelet Denoising        | 64.2    | 28.8    | 46.50        | 2913         | 55.4%              |
> | Split Bregman            | 64.7    | 31.6    | 48.15        | 2345         | 25.1%              |
> | TVM (Guo et al.[1])      | 56.9    | 50.8    | 53.85        | 7983         | 325.8%             |
> | Adaptive-FGP             | 66.1    | 53.0    | 59.55        | 2075         | 10.7%              |
>
> **Table 2. Adaptive-FGP Latency per Image (ms)**
>
> | Max Time | Min Time | Average Time |
> |----------|----------|--------------|
> |   5.72   |   2.81   |     3.71     |
>
>
> **Answer for Q1:**
>
> **Table3. PGD-L∞ vs. PGD-L2 Accuracy (Base-to-Novel, Across Datasets)**
>
> | Dataset        | Type  | EPS1 PGD Linf | EPS1 PGD L2 | EPS2 PGD Linf | EPS2 PGD L2 | EPS4 PGD Linf | EPS4 PGD L2 |
> |----------------|-------|----------------|--------------|----------------|--------------|----------------|--------------|
> | Caltech101     | Base  | 94.38          | 94.25        | 93.35          | 92.32        | 88.32          | 90.70        |
> |                | Novel | 88.03          | 87.99        | 84.83          | 84.06        | 75.11          | 81.00        |
> | DTD            | Base  | 67.98          | 68.29        | 63.77          | 62.38        | 54.75          | 61.00        |
> |                | Novel | 43.88          | 44.44        | 40.34          | 37.80        | 32.97          | 36.96        |
> | EuroSAT        | Base  | 84.67          | 85.45        | 79.00          | 79.64        | 74.52          | 81.00        |
> |                | Novel | 47.40          | 55.23        | 53.36          | 54.49        | 50.56          | 54.38        |
> | FGVCAircraft   | Base  | 25.37          | 25.87        | 20.11          | 20.05        | 15.97          | 18.91        |
> |                | Novel | 16.68          | 16.86        | 14.04          | 12.96        | 10.62          | 13.92        |
> | OxfordPets     | Base  | 78.72          | 80.75        | 69.59          | 73.52        | 54.12          | 70.28        |
> |                | Novel | 83.71          | 83.95        | 73.21          | 75.17        | 57.10          | 70.86        |
>
> **Answer for Q2:**
>
> **Table4. Natural and Robust Performance of CLIP with RN101 and RN50 Backbones under the 16-Shot Setting**
>
> | Dataset     | Metric | RN101 | RN50 |
> |-------------|--------|-------|------|
> | EuroSAT     | Nat    | 67.73 | 68.99 |
> |             | Adv    | 48.33 | 53.23 |
> | OxfordPets  | Nat    | 25.67 | 23.58 |
> |             | Adv    | 18.56 | 16.90 |
>
> **Answer for Q3:**
>
> To further enhance the quality of our work, we extended the evaluation to more complex image understanding tasks. We selected the stronger LLaVA architecture as the evaluation target. Since LLaVA only supports the CLIP (ViT-L/14) backbone, we retrained CLIP (ViT-L/14) on the ImageNet dataset and transferred the adversarial prompts trained on classification tasks to LLaVA's image captioning task. The latent-space alignment of CoAPT effectively captures high-level semantics and demonstrates strong performance across both classification and understanding tasks, where abstraction and semantic comprehension of objects are particularly critical.We evaluate image captioning performance using CIDEr scores on the larger VLM model LLaVA-1.5 7B. The evaluation is conducted on the Flickr30k and COCO2014 datasets. Specifically, during the attack phase, we adopt a FARE[2]-based attack pipeline designed to substantially degrade LLaVA performance while maintaining computational feasibility. The attack process employs five ground truth captions per sample as sequential targets, executing five rounds of half-precision (float16) APGD attacks to maximize effectiveness by leveraging multiple reference annotations. For each sample, the attack pipeline maintains a cumulative best-result dictionary that tracks the strongest degradation achieved for each image or query. Once the performance score falls below a predefined threshold, the corresponding sample is excluded from subsequent attack rounds. The CIDEr threshold is set to 10.0 for the COCO dataset and 2.0 for Flickr30k. The APGD attack is conducted under the $L_\infty$ norm with perturbation budgets of 1/255, 2/255, and 4/255, using 100 iterations. Table 5 provides representative examples of CIDEr score degradation across attack rounds.
>
> Both the base model CLIP and the SOTA method FARE[2] are evaluated under the exact same configuration as CoAPT, using the official pretrained weights provided by the FARE repository. Since FARE only provides models trained with perturbation budgets of EPS=2 and EPS=4, we currently report its performance under these settings only. At present, evaluations are conducted in half precision (float16) on 100 randomly sampled images with a fixed seed to balance computational efficiency and attack effectiveness. In the final version of the paper, we plan to extend the evaluation to 500–1000 images (as FARE reports results on 500) and include an additional attack round in single precision (float32). If FARE exhibits favorable adaptation at EPS=1, we will also report its performance under this setting.
>
> As shown in Table 6 and Table 7, CoAPT consistently achieves superior clean and robust accuracy across all EPS settings. Compared to the FARE method, CoAPT delivers an average improvement of 36.15 (EPS2) and 40.465 (EPS4) in robust CIDEr scores across two datasets, while maintaining higher performance on clean data. Our method employs 20 rounds of 16-shot training on ImageNet, using only 16,000 training images, which is significantly fewer in both data volume and training time than the full dual-pass ImageNet training adopted by FARE.
>
> **Table 5. CIDEr Degradation across Attack Rounds and EPS**
>
> | Pert. budg.     | Vanilla CLIP EPS2 | Vanilla CLIP EPS4 | CoAPT EPS2 | CoAPT EPS4 |
> |-----------------|-------------------|--------------------|------------|------------|
> | Nat             | 81.51             | 81.51              | 82.89      | 85.04      |
> | APGD with GTC 1 | 12.81             | 8.39               | 76.91      | 69.58      |
> | APGD with GTC 2 | 7.86              | 3.81               | 72.33      | 64.13      |
> | APGD with GTC 3 | 6.55              | 2.73               | 70.06      | 62.42      |
> | APGD with GTC 4 | 4.22              | 1.86               | 68.63      | 61.18      |
> | APGD with GTC 5 | 3.52              | 1.71               | 67.30      | 59.00      |
>
>
> **Table 6. Clean Performance of LLaVA Variants with Different Visual Encoders on Image Captioning**
>
> | Method | Flickr30K EPS1 | EPS2 | EPS4 | COCO EPS1 | EPS2 | EPS4 |
> |--------|----------------|------|------|------------|------|------|
> | CLIP   | 81.51          | 81.51| 81.51| 125.81     |125.81|125.91|
> | FARE   | -              | 80.05| 75.78| -          |115.45|110.11|
> | CoAPT  | 84.23          | 82.89| 85.04| 119.61     |128.94|120.69|
>
> **Table 7. Robust Performance of LLaVA Variants with Different Visual Encoders on Image Captioning**
>
> | Method | Flickr30K EPS1 | EPS2 | EPS4 | COCO EPS1 | EPS2  | EPS4  |
> |--------|----------------|------|------|-----------|-------|--------|
> | CLIP   | 9.12           | 3.52 | 1.71 | 10.93     | 4.28  | 3.04   |
> | FARE   | -              | 40.71| 32.95| -         | 60.82 | 42.03  |
> | CoAPT  | 73.27          | 67.30| 59.00| 107.74    |106.53 | 96.91  |
>
>
> [1]Guo, C., et al. Countering adversarial images using input transformations. ICLR 2018.
> [2]Schlarmann, C., et al. Robust CLIP: Unsupervised adversarial fine-tuning of vision embeddings for robust large vision-language models. ICML2024.

---

### Official Review · Reviewer_rp8m · 2025-07-01

**Clarity:** 2
**Significance:** 3
**Originality:** 3
**Rating:** 5
**Confidence:** 3

**Summary:**

This paper proposes an adversarial prompt tuning method that leverages the embeddings of natural images and raw text to guide the optimization of adversarial prompts. It also introduces a real-time total variation regularization technique to suppress adversarial perturbations in high-frequency features. Additionally, the paper presents a regularization strategy to reduce the divergence in prediction distributions between natural and adversarial samples. The results demonstrated across multiple datasets and benchmarks show significant performance improvements.

**Questions:**

1. The paper only investigates image classification tasks. Can the adversarial prompts trained on classification tasks be transferred to other tasks, such as image captioning?
2. If the goal is merely to remove high-frequency perturbations from adversarial examples, could methods like HGD be used as an alternative to adaptive-FGP?

**Ethical Concerns:**

["NO or VERY MINOR ethics concerns only"]

**Final Justification:**

I believe this paper can make contribution to the research field of multimodal robustness, so I recommend acceptance.

**Limitations:**

yes

**Quality:**

3

**Strengths And Weaknesses:**

Strengths:
1. The proposed method is very lightweight, yet simple and effective.
2. It has been tested on diverse datasets, with experimental results showing significant improvements.
3. Comprehensive and detailed ablation studies demonstrate the contribution of each component of the method.

Weaknesses:
1. The paper includes an excessive amount of background information, which results in many important technical details (such as Algorithm 2) being relegated to the supplementary material.

---

> ### Author Rebuttal · Authors · 2025-07-31
>
> **We thank reviewer for the constructive comments. We provide our feedback as follows. This will ultimately be reflected in the revision of the paper.**
>
> **Answer for W1:**
> We will revise the paper structure by condensing the background section and incorporating key technical details, such as Algorithm 2, into the main text.
>
>
> **Answer for Q1:**
>
> We comprehensively evaluated both task-specific and generalized image classification tasks across four standard benchmarks. To further enhance the quality of our work, we extended the evaluation to more complex image understanding tasks. We selected the stronger LLaVA architecture as the evaluation target. Since LLaVA only supports the CLIP (ViT-L/14) backbone, we retrained CLIP (ViT-L/14) on the ImageNet dataset and transferred the adversarial prompts trained on classification tasks to LLaVA's image captioning task. The latent-space alignment of CoAPT effectively captures high-level semantics and demonstrates strong performance across both classification and understanding tasks, where abstraction and semantic comprehension of objects are particularly critical.We evaluate image captioning performance using CIDEr scores on the larger VLM model LLaVA-1.5 7B. The evaluation is conducted on the Flickr30k and COCO2014 datasets. Specifically, during the attack phase, we adopt a FARE[1]-based attack pipeline designed to substantially degrade LLaVA performance while maintaining computational feasibility. The attack process employs five ground truth captions per sample as sequential targets, executing five rounds of half-precision (float16) APGD attacks to maximize effectiveness by leveraging multiple reference annotations. For each sample, the attack pipeline maintains a cumulative best-result dictionary that tracks the strongest degradation achieved for each image or query. Once the performance score falls below a predefined threshold, the corresponding sample is excluded from subsequent attack rounds. The CIDEr threshold is set to 10.0 for the COCO dataset and 2.0 for Flickr30k. The APGD attack is conducted under the $L_\infty$ norm with perturbation budgets of 1/255, 2/255, and 4/255, using 100 iterations. Table 1 provides representative examples of CIDEr score degradation across attack rounds.
>
> Both the base model CLIP and the SOTA method FARE[1] are evaluated under the exact same configuration as CoAPT, using the official pretrained weights provided by the FARE repository. Since FARE only provides models trained with perturbation budgets of EPS=2 and EPS=4, we currently report its performance under these settings only. At present, evaluations are conducted in half precision (float16) on 100 randomly sampled images with a fixed seed to balance computational efficiency and attack effectiveness. In the final version of the paper, we plan to extend the evaluation to 500–1000 images (as FARE reports results on 500) and include an additional attack round in single precision (float32). If FARE exhibits favorable adaptation at EPS=1, we will also report its performance under this setting.
>
> As shown in Table 2 and Table 3, CoAPT consistently achieves superior clean and robust accuracy across all EPS settings. Compared to the FARE method, CoAPT delivers an average improvement of 36.15 (EPS2) and 40.465 (EPS4) in robust CIDEr scores across two datasets, while maintaining higher performance on clean data. Our method employs 20 rounds of 16-shot training on ImageNet, using only 16,000 training images, which is significantly fewer in both data volume and training time than the full dual-pass ImageNet training adopted by FARE.
>
> **Table 1. CIDEr Degradation across Attack Rounds and EPS**
>
> | Pert. budg.     | Vanilla CLIP EPS2 | Vanilla CLIP EPS4 | CoAPT EPS2 | CoAPT EPS4 |
> |-----------------|-------------------|--------------------|------------|------------|
> | Nat             | 81.51             | 81.51              | 82.89      | 85.04      |
> | APGD with GTC 1 | 12.81             | 8.39               | 76.91      | 69.58      |
> | APGD with GTC 2 | 7.86              | 3.81               | 72.33      | 64.13      |
> | APGD with GTC 3 | 6.55              | 2.73               | 70.06      | 62.42      |
> | APGD with GTC 4 | 4.22              | 1.86               | 68.63      | 61.18      |
> | APGD with GTC 5 | 3.52              | 1.71               | 67.30      | 59.00      |
>
>
> **Table 2. Clean Performance of LLaVA Variants with Different Visual Encoders on Image Captioning**
>
> | Method | Flickr30K EPS1 | EPS2 | EPS4 | COCO EPS1 | EPS2 | EPS4 |
> |--------|----------------|------|------|------------|------|------|
> | CLIP   | 81.51          | 81.51| 81.51| 125.81     |125.81|125.91|
> | FARE   | -              | 80.05| 75.78| -          |115.45|110.11|
> | CoAPT  | 84.23          | 82.89| 85.04| 119.61     |128.94|120.69|
>
> **Table 3. Robust Performance of LLaVA Variants with Different Visual Encoders on Image Captioning**
>
> | Method | Flickr30K EPS1 | EPS2 | EPS4 | COCO EPS1 | EPS2  | EPS4  |
> |--------|----------------|------|------|-----------|-------|--------|
> | CLIP   | 9.12           | 3.52 | 1.71 | 10.93     | 4.28  | 3.04   |
> | FARE   | -              | 40.71| 32.95| -         | 60.82 | 42.03  |
> | CoAPT  | 73.27          | 67.30| 59.00| 107.74    |106.53 | 96.91  |
>
>
> **Answer for Q2:**
> The HGD method should work, but we found that some common denoising methods do not work well. Specifically,we focus on denoising methods with real-time capability and evaluate several classical approaches in terms of both performance and computational efficiency. As shown in Table 4, we compare the total runtime for 16-shot training and testing on the DTD dataset using an NVIDIA Tesla V100 GPU. The proposed adaptive-FGP method employs 10 iterative steps. Under this setting, the time overhead increases by only 10.7\% compared to the baseline model without contrastive mechanisms. Given the observed performance gains, this overhead remains acceptable. For other classical baselines, wavelet denoising and the Split Bregman method are implemented via the official skimage library, while TVM adopts a precompiled C++ dynamic library for Total Variation denoising. Notably, our adaptive-FGP method incorporates a parallelized implementation that runs directly on CUDA, leading to significantly lower runtime compared to others. However, these runtime differences do not directly reflect computational complexity. Table 5 further reports the average, minimum, and maximum per-image inference time of adaptive-FGP, measured after one epoch over the entire training set.
>
> In terms of accuracy, our method achieves state-of-the-art performance on both clean and robust evaluations,  particularly excelling in robustness. By integrating a spatially adaptive regularization strategy and an accelerated gradient method with an adaptive restart mechanism, we improve the traditional TV approach to simultaneously enhance performance and real-time capability.
>
> **Table 4. Denoising Runtime and Accuracy (Natural vs. Adversarial)**
>
> | Method                    | Nat (%) | Rob (%) | Average (%) | Runtime (s) | Time Increase (%) |
> |--------------------------|---------|---------|--------------|--------------|--------------------|
> | No Denoising  | 64.3    | 27.8    | 46.05        | 1875         | ~                  |
> | Wavelet Denoising        | 64.2    | 28.8    | 46.50        | 2913         | 55.4%              |
> | Split Bregman            | 64.7    | 31.6    | 48.15        | 2345         | 25.1%              |
> | TVM (Guo et al.[2])      | 56.9    | 50.8    | 53.85        | 7983         | 325.8%             |
> | Adaptive-FGP             | 66.1    | 53.0    | 59.55        | 2075         | 10.7%              |
>
> **Table 5. Adaptive-FGP Latency per Image (ms)**
>
> | Max Time | Min Time | Average Time |
> |----------|----------|--------------|
> |   5.72   |   2.81   |     3.71     |
>
> [1]Schlarmann, C., et al. Robust CLIP: Unsupervised adversarial fine-tuning of vision embeddings for robust large vision-language models. ICML2024.
> [2]Guo, C., et al. Countering adversarial images using input transformations. ICLR 2018.

---

> > ### Comment · Reviewer_rp8m · 2025-08-04
> >
> > I believe the authors' rebuttal has addressed most of my concerns. I sincerely appreciate their efforts in this regard. I will maintain my score and recommend the acceptance of the paper.

---

> > > ### Author Response · Authors · 2025-08-04
> > >
> > > Thank you for your positive follow-up and thoughtful assessment. We are glad that our rebuttal has successfully addressed your concerns. We sincerely appreciate your recognition of our work and efforts on this paper.

---

### Official Review · Reviewer_e7hZ · 2025-07-03

**Clarity:** 3
**Significance:** 2
**Originality:** 2
**Rating:** 4
**Confidence:** 4

**Summary:**

This paper proposes CoAPT, a novel method to enhance the adversarial robustness of Vision-Language Models (VLMs) while preserving their natural generalization capabilities. CoAPT employs an improved real-time total variation technique to suppress high-frequency details in input images, compressing the adversarial perturbation space. It then reconstructs corrupted natural features in the latent space of pre-trained VLMs, guided by high-level image and text representations. The approach achieves a trade-off between robustness and generalization, outperforming SOTA methods across four benchmark tasks (few-shot, base-to-novel, zero-shot, and out-of-distribution generalization).

**Questions:**

1. The paper should more explicitly differentiate CoAPT from prior work using latent-space alignment and TV regularization.

2. The paper would benefit from adding theoretical analysis explaining why latent-space reconstruction better preserves generalization compared to alternatives.

3. While the CLIP results are compelling, testing on at least one additional VLM architecture (e.g., LLaVA or BLIP) would significantly strengthen claims about general applicability.

**Ethical Concerns:**

["NO or VERY MINOR ethics concerns only"]

**Final Justification:**

I tend to increase my rating.

**Limitations:**

It is recommended to discusses computational costs, generalization beyond CLIP, and potential misuse risks (e.g., stronger adversarial attacks).

**Quality:**

3

**Strengths And Weaknesses:**

Strengths:
1. Extensive experiments are conducted across 15 datasets and four benchmark tasks. The results show consistent improvements in both robustness and generalization, with detailed ablation studies justifying design choices.

2. The paper is well-structured, with a clear problem statement, methodology, and experimental setup. The figures and tables effectively summarize key results.

Weaknesses:
1. The core idea of latent-space alignment has appeared in prior work (e.g., in self-supervised learning), and the use of TV regularization for adversarial defense is not entirely new. While the integration of these components is novel, the paper could better delineate how CoAPT differs from related approaches in robust fine-tuning and adversarial prompting.

2. Some technical details (e.g., the exact formulation of the low-rank residual module) are briefly mentioned but not deeply explained. A more intuitive breakdown of how Renyi divergence helps compared to standard KL divergence could improve accessibility.

3. The evaluation is limited to CLIP-based models (ViT-B/32). Testing on larger VLMs (e.g., LLaVA, Flamingo) or different architectures would strengthen claims about general applicability.

4. While the empirical results are strong, the paper does not provide a theoretical analysis of why the proposed latent-space reconstruction preserves generalization better than alternatives. Additionally, the computational cost of adaptive-FGP is not thoroughly discussed—real-time performance claims should be quantified with latency benchmarks.

---

> ### Author Rebuttal · Authors · 2025-07-30
>
> **We thank reviewer for the constructive comments. We provide our feedback as follows.**
>
> **Answer for W1&Q1:**
> Guo et al. [1] did not optimize TV regularization for generalization, performance, or efficiency. Their method significantly degraded clean features and required costly offline denoising due to high computational overhead. In contrast, CoAPT restores natural features via latent-space reconstruction with only ~10% added training cost.
>
> Unimodal visual- or text-only prompting lacks cross-modal guidance and fails to leverage the joint benefits of multimodal prompting against adversarial samples. While FAP [2] introduced multimodal prompts, it relied solely on comparing logits from robust CLIP models, limiting generalization. In contrast, CoAPT uses latent features from the original CLIP for training, differing fundamentally in both design and implementation.
>
> Fare [3] conducts full-parameter fine-tuning of the visual encoder via latent-space alignment, incurring high training costs and overfitting risks that weaken generalization. It also overlooks the role of the text encoder in resisting visual adversarial noise. In contrast, CoAPT applies lightweight adversarial prompt modulation, preserving the pre-trained VLM's generalization while supporting longer training. It further leverages the complementary effect of text prompts and explores synergy between textual and visual cues, advancing multimodal adversarial defense.
>
> **Answer for W2:**
>
> Due to space limitations, we will add more details to the LoRA residual module and Rényi loss in the final version.
>
>
> **Answer for W3&Q3:**
>
> We provide additional image classification results based on CLIP (ViT-B/16) model in the appendix. Furthermore, to assess the general applicability of our approach, we evaluate image captioning performance using CIDEr scores on the larger VLM model LLaVA-1.5 7B. The evaluation is conducted on the Flickr30k and COCO2014 datasets. Specifically, during the attack phase, we adopt a FARE[3]-based attack pipeline designed to substantially degrade LLaVA performance while maintaining computational feasibility. The attack process employs five ground truth captions per sample as sequential targets, executing five rounds of half-precision (float16) APGD attacks to maximize effectiveness by leveraging multiple reference annotations. For each sample, the attack pipeline maintains a cumulative best-result dictionary that tracks the strongest degradation achieved for each image or query. Once the performance score falls below a predefined threshold, the corresponding sample is excluded from subsequent attack rounds. The CIDEr threshold is set to 10.0 for the COCO dataset and 2.0 for Flickr30k. The APGD attack is conducted under the $L_\infty$ norm with perturbation budgets of 1/255, 2/255, and 4/255, using 100 iterations. Table 1 provides representative examples of CIDEr score degradation across attack rounds.
>
> Both the base model CLIP and the SOTA method FARE[3] are evaluated under the exact same configuration as CoAPT, using the official pretrained weights provided by the FARE repository. Since FARE only provides models trained with perturbation budgets of EPS=2 and EPS=4, we currently report its performance under these settings only.
>
> As shown in Table 2 and Table 3, CoAPT consistently achieves superior clean and robust accuracy across all EPS settings. Compared to the FARE method, CoAPT delivers an average improvement of 36.15 (EPS2) and 40.465 (EPS4) in robust CIDEr scores across two datasets, while maintaining higher performance on clean data. Our method employs 20 rounds of 16-shot training on ImageNet, using only 16,000 training images, which is significantly fewer in both data volume and training time than the full dual-pass ImageNet training adopted by FARE.
>
> **Table 1. CIDEr Degradation across Attack Rounds and EPS**
>
> | Pert. budg.     | Vanilla CLIP EPS2 | Vanilla CLIP EPS4 | CoAPT EPS2 | CoAPT EPS4 |
> |-----------------|-------------------|--------------------|------------|------------|
> | Nat             | 81.51             | 81.51              | 82.89      | 85.04      |
> | APGD with GTC 1 | 12.81             | 8.39               | 76.91      | 69.58      |
> | APGD with GTC 2 | 7.86              | 3.81               | 72.33      | 64.13      |
> | APGD with GTC 3 | 6.55              | 2.73               | 70.06      | 62.42      |
> | APGD with GTC 4 | 4.22              | 1.86               | 68.63      | 61.18      |
> | APGD with GTC 5 | 3.52              | 1.71               | 67.30      | 59.00      |
>
>
> **Table 2. Clean Performance of LLaVA Variants with Different Visual Encoders on Image Captioning**
>
> | Method | Flickr30K EPS1 | EPS2 | EPS4 | COCO EPS1 | EPS2 | EPS4 |
> |--------|----------------|------|------|------------|------|------|
> | CLIP   | 81.51          | 81.51| 81.51| 125.81     |125.81|125.91|
> | FARE   | -              | 80.05| 75.78| -          |115.45|110.11|
> | CoAPT  | 84.23          | 82.89| 85.04| 119.61     |128.94|120.69|
>
> **Table 3. Robust Performance of LLaVA Variants with Different Visual Encoders on Image Captioning**
>
> | Method | Flickr30K EPS1 | EPS2 | EPS4 | COCO EPS1 | EPS2  | EPS4  |
> |--------|----------------|------|------|-----------|-------|--------|
> | CLIP   | 9.12           | 3.52 | 1.71 | 10.93     | 4.28  | 3.04   |
> | FARE   | -              | 40.71| 32.95| -         | 60.82 | 42.03  |
> | CoAPT  | 73.27          | 67.30| 59.00| 107.74    |106.53 | 96.91  |
>
>
>
> **Answer for W4&Q2:**
> In the context of representation learning, the latent space has been theoretically substantiated as a more semantically aligned modeling domain than the raw input space. Moschella et al. [6] empirically demonstrated that the relative geometric configurations among latent representations, such as angular distributions, remain highly consistent under different initialization and training trajectories, indicating the training stability of the latent space and enabling cross-task and cross-modal generalization through inter-space communication. Building on this, Dou et al. [7] construct class-disentangled, near-orthogonal latent subspaces and employ Gram matrix-based structural analysis to reveal a quantitative association between subspace geometry and robust generalization error, theoretically reinforcing the link between latent structural regularity and generalization robustness. These insights collectively support latent-space reconstruction not as an ad hoc empirical choice, but as a theoretically motivated modeling strategy, wherein the associated inductive bias plays a pivotal role in improving generalization in the target task.
>
>
>
> We focus on denoising methods with real-time capability and evaluate several classical approaches in terms of both performance and computational efficiency. As shown in Table 4, we compare the total runtime for 16-shot training and testing on the DTD dataset using an NVIDIA Tesla V100 GPU. The proposed adaptive-FGP method employs 10 iterative steps. Under this setting, the time overhead increases by only 10.7\% compared to the baseline model without contrastive mechanisms. Given the observed performance gains, this overhead remains acceptable. For other classical baselines, wavelet denoising and the Split Bregman method are implemented via the official skimage library, while TVM adopts a precompiled C++ dynamic library for Total Variation denoising. Notably, our adaptive-FGP method incorporates a parallelized implementation that runs directly on CUDA, leading to significantly lower runtime compared to others. However, these runtime differences do not directly reflect computational complexity. Table 5 further reports the average, minimum, and maximum per-image inference time of adaptive-FGP, measured after one epoch over the entire training set.
>
> In terms of accuracy, our method achieves state-of-the-art performance on both clean and robust evaluations,  particularly excelling in robustness. By integrating a spatially adaptive regularization strategy and an accelerated gradient method with an adaptive restart mechanism, we improve the traditional TV approach to simultaneously enhance performance and real-time capability.
>
> **Table 4. Denoising Runtime and Accuracy (Natural vs. Adversarial)**
>
> | Method                    | Nat (%) | Rob (%) | Average (%) | Runtime (s) | Time Increase (%) |
> |--------------------------|---------|---------|--------------|--------------|--------------------|
> | No Denoising  | 64.3    | 27.8    | 46.05        | 1875         | ~                  |
> | Wavelet Denoising        | 64.2    | 28.8    | 46.50        | 2913         | 55.4%              |
> | Split Bregman            | 64.7    | 31.6    | 48.15        | 2345         | 25.1%              |
> | TVM (Guo et al.[1])      | 56.9    | 50.8    | 53.85        | 7983         | 325.8%             |
> | Adaptive-FGP             | 66.1    | 53.0    | 59.55        | 2075         | 10.7%              |
>
> **Table 5. Adaptive-FGP Latency per Image (ms)**
>
> | Max Time | Min Time | Average Time |
> |----------|----------|--------------|
> |   5.72   |   2.81   |     3.71     |
>
>
>
> [1]Countering adversarial images using input transformations. ICLR 2018.
> [2]Few-shot adversarial prompt learning on vision-language models. NeurIPS 2024.
> [3]Robust CLIP: Unsupervised adversarial fine-tuning of vision embeddings for robust large vision-language models. ICML2024.
> [4]α-GAN: Convergence and estimation guarantees. ISIT2022.
> [5]A tunable loss function for robust classification: Calibration, landscape, and generalization.IEEE TIT2022.
> [6]Relative Representations Enable Zero-Shot Latent Space Communication. ICLR2023.
> [7]Improving Robust Generalization with Diverging Spanned Latent Space. TMLR2025.

---

> > ### Comment · Reviewer_e7hZ · 2025-08-07
> >
> > Thank you for the detailed and thoughtful rebuttal. The authors have provided substantial clarifications and new experimental evidence that address most of my concerns. In particular, I appreciate the additional evaluations on CLIP ViT-B/16 and LLaVA-1.5, which strengthen the generality claims. The discussion on latent-space reconstruction is now better supported by theoretical insights and references, and the efficiency analysis of the adaptive-FGP method is well-presented. While some components (e.g., low-rank residual modules and Rényi divergence) could still benefit from further elaboration or visualization, the current version reflects a clear improvement in both methodological soundness and empirical validation. Based on these improvements, I am willing to revise my score.

---

> > > ### Author Response · Authors · 2025-08-08
> > >
> > > We appreciate your positive feedback and detailed point-by-point response. Your comments on latent space reconstruction, efficiency, and the proposal to experiment on LLaVA provide valuable perspectives for enhancing the generalizability of our approach and improving the overall quality of the paper. We are pleased that our rebuttal has addressed your concerns, and we will incorporate your suggestions to further refine our manuscript.

---

### Comment · Area_Chair_QaTm · 2025-08-04

Dear Reviewers,

The deadline for author-reviewer discussion period (August 6, AoE) is approaching soon, but some reviewers have not provided responses to the authors. Your feedback/response to the authors' rebuttal is highly expected.

Best,
AC

---

### Decision · Program_Chairs · 2025-09-17

**Decision:**

Accept (spotlight)

**Comment:**

The paper proposes Collaborative Adversarial Prompt Tuning (CoAPT) to enhance the adversarial robustness of VLMs in the latent space. The paper is clearly written and well-structured, presenting strong performance across multiple datasets and benchmarks. It makes a valuable contribution to the research field of the multimodal community. The authors’ rebuttal successfully addressed some concerns regarding practical overhead, generalizability, and additional experimental evaluations. These revisions should be incorporated into the final version.